# Bacterial capsular polysaccharides with antibiofilm activity share common biophysical and electrokinetic properties

Joaquín Bernal-Bayard [1,2], Jérôme Thiebaud[3], Marina Brossaud[3], Audrey Beaussart [4], Céline Caillet [4], Yves Waldvogel[4], Laetitia Travier[1,5], Sylvie Létoffé[1], Thierry Fontaine[6], Bachra Rokbi[3], Philippe Talaga[3], Christophe Beloin [1], Noëlle Mistretta [3] ✉, Jérôme F. L. Duval [4] ✉ & Jean-Marc Ghigo [1] ✉

Bacterial biofilms are surface-attached communities that are difficult to eradicate due to a high tolerance to antimicrobial agents. The use of non-biocidal surface-active compounds to prevent the initial adhesion and aggregation of bacterial pathogens is a promising alternative to antibiotic treatments and several antibiofilm compounds have been identified, including some capsular polysaccharides released by various bacteria. However, the lack of chemical and mechanistic understanding of the activity of these polymers limits their use to control biofilm formation. Here, we screen a collection of 31 purified capsular polysaccharides and first identify seven new compounds with non-biocidal activity against *Escherichia coli* and/or *Staphylococcus aureus* biofilms. We measure and theoretically interpret the electrophoretic mobility of a subset of 21 capsular polysaccharides under applied electric field conditions, and we show that active and inactive polysaccharide polymers display distinct electrokinetic properties and that all active macromolecules share high intrinsic viscosity features. Despite the lack of specific molecular motif associated with antibiofilm properties, the use of criteria including high density of electrostatic charges and permeability to fluid flow enables us to identify two additional capsular polysaccharides with broad-spectrum antibiofilm activity. Our study therefore provides insights into key biophysical properties discriminating active from inactive polysaccharides. The characterization of a distinct electrokinetic signature associated with antibiofilm activity opens new perspectives to identify or engineer non-biocidal surface-active macromolecules to control biofilm formation in medical and industrial settings.

Bacterial biofilms are widespread surface-attached or aggregated bacteria that can negatively impact human activities when developing on medical or industrial surfaces[1,2]. Due to their high tolerance to antibiotics, biofilms are difficult to eradicate, and the prevention of biofilm-associated infections is a major health and economic issue[3,4]. Strategies to prevent biofilm formation often target the initial steps of bacterial adhesion using surfaces coated by biocidal agents such as broad-spectrum antibiotics or heavy metals[5]. These biocidal

approaches are limited by the rapid accumulation of dead bacteria and organic debris, which reduces the activity of the coated surfaces toward new incoming cells. Moreover, the use of surfaces releasing biocides such as antibiotics is associated with worrisome selection of antibiotic resistance[6].

Several studies have shown that non-antibiotic anti-adhesion strategies could also efficiently interfere with bacterial biofilm formation[7–14]. The design of bio-inspired materials with anti-adhesion surface properties has been proposed to constitute an effective solution to protect patient care equipment from pathogen colonization, and therefore impede key steps of infection, from initial surface contact to subsequent bacteria-bacteria interactions[15–17]. Non-biocidal and bio-based strategies are also actively explored, including approaches preventing and/or disrupting biofilms using quorum sensing inhibitors that interfere with bacterial communications[18]. Bacteria also secrete biosurfactants altering material surface properties such as wettability and charge[19,20]. These surface-active compounds reduce surface contacts and contribute to bacterial motility or are involved in competitive interactions between bacteria[12]. Whereas many of these molecules correspond to small lipopeptides, recent studies showed that high molecular weight capsular polysaccharides released by various bacteria could prevent adhesion leading to subsequent cell aggregation and formation of biofilm by a wide range of Gram+ and Gram- bacteria. These include several nosocomial pathogens such as *Escherichia coli*, *Pseudomonas aeruginosa*, *Klebsiella pneumoniae*, *Staphylococcus aureus*, *Staphylococcus epidermidis*, and *Enterococcus faecalis*[7,9,11,21–24].

Unlike secreted bacterial antagonistic macromolecules such as colicins, toxins, phages, and a few toxic surface-active compounds[25,26], these antibiofilm polysaccharides are non-biocidal[11]. They impair bacteria-biotic/abiotic surface interactions mediated by adhesion factors such as pili, adhesins or extracellular matrix polymers via the modification of the wettability, charge, and overall bacteria-surface contact properties[7,12,18]. The use of these non-biocidal antibiofilm macromolecules was proposed as a promising resistance-free approach to reduce pathogenic biofilms and biofilm-associated bacterial infections while avoiding side effects caused by broad-spectrum biocides[11,23,27]. However, the limited description of the chemical and structural bases of the activity of these antibiofilm macromolecules severely hinders their prophylactic use for bacterial biofilm control.

Here, we investigate the antibiofilm activity of a panel of 31 purified Gram+ or Gram bacterial capsular polysaccharides of known composition and structure. Among those, we found nine new non-biocidal polysaccharides inhibiting biofilm formation by prototypical nosocomial pathogens, including *E. coli* and *S. aureus*. This enabled us to perform a structure-function comparison of enough active and inactive polysaccharides and to show that antibiofilm capsular polysaccharides are characterized by a high intrinsic viscosity and a specific electrokinetic signature. Our study, therefore, identifies key chemical and biophysical properties of bacterial antibiofilm polysaccharides, thus providing insights that pave the way for the engineering of new and well-defined non-biocidal surface-active macromolecules to control biofilm formation in medical and industrial settings.

## Results

### Size integrity is a key parameter of antibiofilm polysaccharide activity

To investigate the potential relationships between composition, structure, size, and activity of non-biocidal bacterial antibiofilm polysaccharides, we first determined the structure of Group 2 capsule (G2cps), a previously identified hydrophilic and negatively charged antiadhesion polysaccharide active on both Gram+ and Gram- bacteria, which is naturally produced and released, notably, by uro-pathogenic *E. coli* strains[7]. G2cps was analyzed by High-Performance

Anion-Exchange Chromatography – Pulsed Amperometric Detection (HPAEC-PAD), High Performance Size-Exclusion Chromatography coupled to Static Light Scattering (HPSEC-LS) and by [1]H, [31]P and [13]C nuclear magnetic resonance (NMR) studies. These analyzes showed that G2cps polysaccharide consists of repeating units composed of O-acetylated and glycerol phosphate residues with an average molecular weight of 800 kDa, as illustrated in Fig. 1A. This structure is similar to that of *E. coli* polysaccharides K2 and K62 (also named K2ab)[28]. We then gradually reduced the size of G2cps polysaccharide while preserving its structural integrity using radical oxidation hydrolysis (Supplementary Fig. 1). The determination of antibiofilm activity of full length and fragmented G2cps showed that even minor reduction of polysaccharide size resulted in a loss of G2cps activity (Fig. 1B), indicating that the conservation of the size of the G2cps polymer is critical for its antiadhesion properties.

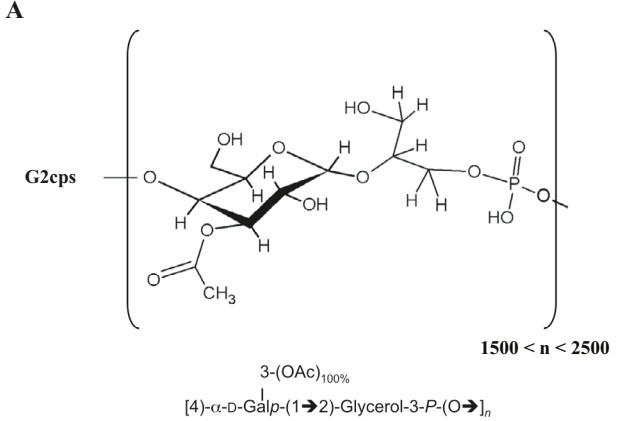

**A**

G2cps

$1500 < n < 2500$

3-(OAc)$_{100\%}$
|
[4)-α-D-Gal*p*-(1➔2)-Glycerol-3-*P*-(O➔]$_n$

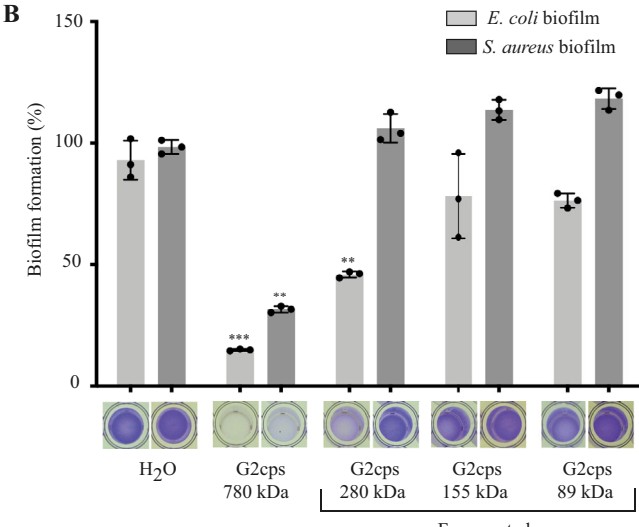

**B**

**Fig. 1 | Structure of the G2cps polysaccharide and its antibiofilm activity as a function of the extent of its fragmentation. A** Structure and composition of G2cps polysaccharide. **B** Biofilm inhibition activity against *E. coli* and *S. aureus* biofilm of native and fragmented G2cps polysaccharide. Biofilm assays were performed in the presence of 50 µg/ml of polysaccharide. Each experiment corresponds to n = 3 independent experiments. Source data are provided as a Source Data file. Statistical analyzes correspond to two-tailed unpaired *t-test* with Welch correction. **\*\*$p < 0.01$; \*\*\*$p < 0.001$. Error bars represent SD.

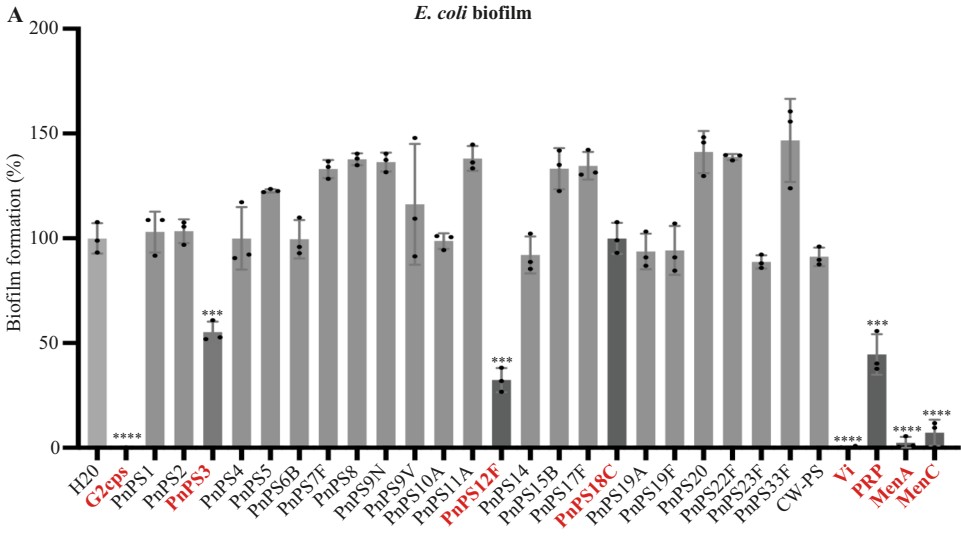

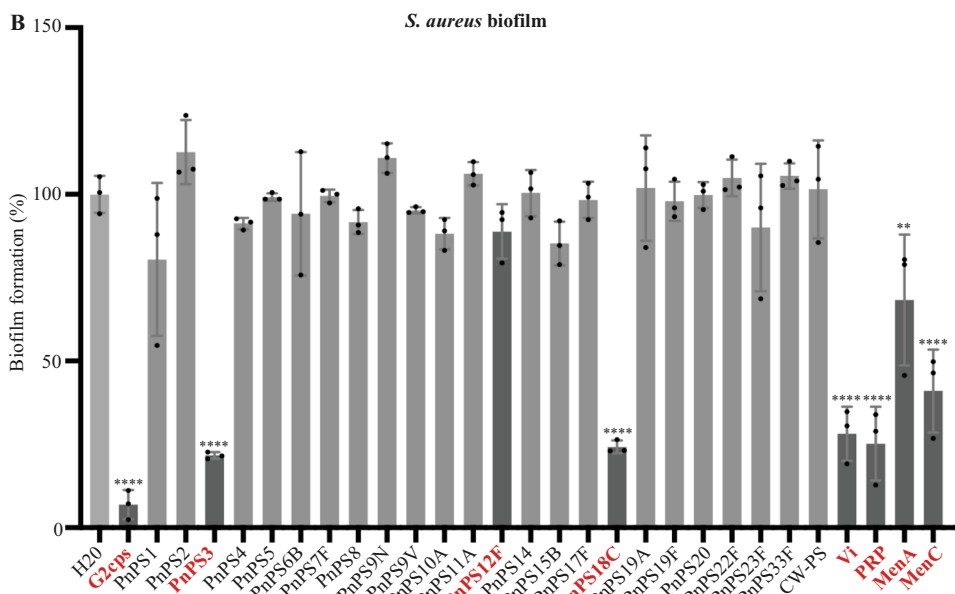

**Fig. 2 | Antibiofilm activity of a collection of bacterial polysaccharides.** *E. coli* (panel **A**) or *S. aureus* (panel **B**) biofilm formation assay in the presence of purified bacterial polysaccharides. G2cps was included as a positive control of active macromolecules and water was used as the negative control. Red: active polysaccharides with Vi, MenA, MenC, G2cps, PRP, and PnPS3 displaying broad spectrum of action, while PnPS18C is only active against Gram+ bacteria and PnPS12F against Gram- bacteria. Each experiment corresponds to $n = 3$ biologically independent experiments in presence of 100 µg/mL of each macromolecule, except for the control ($n = 4$). Source data are provided as a Source Data file. Statistical analyzes correspond to two-tailed unpaired $t$-test with Welch correction. Error bars represent SD. $**p < 0.01$; $***p < 0.001$; $****p < 0.0001$.

## Screening a collection of bacterial capsular polysaccharides reveals compounds with non-biocidal antibiofilm activity

To further attempt to identify G2cps structural or composition features associated with its activity, we screened a collection of 29 different high molecular-weight bacterial polysaccharides to detect macromolecules with G2cps-like antibiofilm properties. Most of them are capsular polysaccharides of known origin, composition, and structure produced and purified from different strains of *Streptococcus pneumoniae*, *Salmonella enterica* serovar Typhi, *Haemophilus influenzae*, *Neisseria meningitidis*, and used as antigens in several polysaccharide and glycoconjugate human vaccines (Supplementary Table 1). Using static microtiter plate biofilm assay followed by crystal violet staining, we showed that, at equivalent concentration (100 µg/ mL), Vi, MenA, and MenC polysaccharides were as active as G2cps in inhibiting *E. coli* biofilm formation but were less active on *S. aureus*

biofilms (Fig. 2A, B). Moreover, whereas PRP and PnPS3 were active on both bacteria, the activity of PnPS12F was restricted to *E. coli* and that of PnPS18C to *S. aureus* (Fig. 2A, B). The comparison of the activity of Vi, PRP, PnPS3, MenA and MenC showed that Vi is the most active polysaccharide and all 5 polysaccharides are non-biocidal (Supplementary Figs. 2 and 3). We confirmed these results with a dynamic assay using continuous flow biofilm microfermentors and showed that Vi also strongly inhibited *E. coli* and *S. aureus* biofilm formation, while this inhibition was not observed with treatment by the inactive polysaccharide PnPS8 (Fig. 3). This analysis also confirmed the narrow-spectrum activity of PnPS18C polysaccharide (active on *S. aureus* but inactive on *E. coli*) and of PnPS12F (inactive on *S. aureus* but active on *E. coli*) (Fig. 3).

Finally, we confirmed the activity of Vi, MenA, MenC, PRP, and PnPS3 on a panel of biofilm-forming Gram+ or Gram- bacteria,

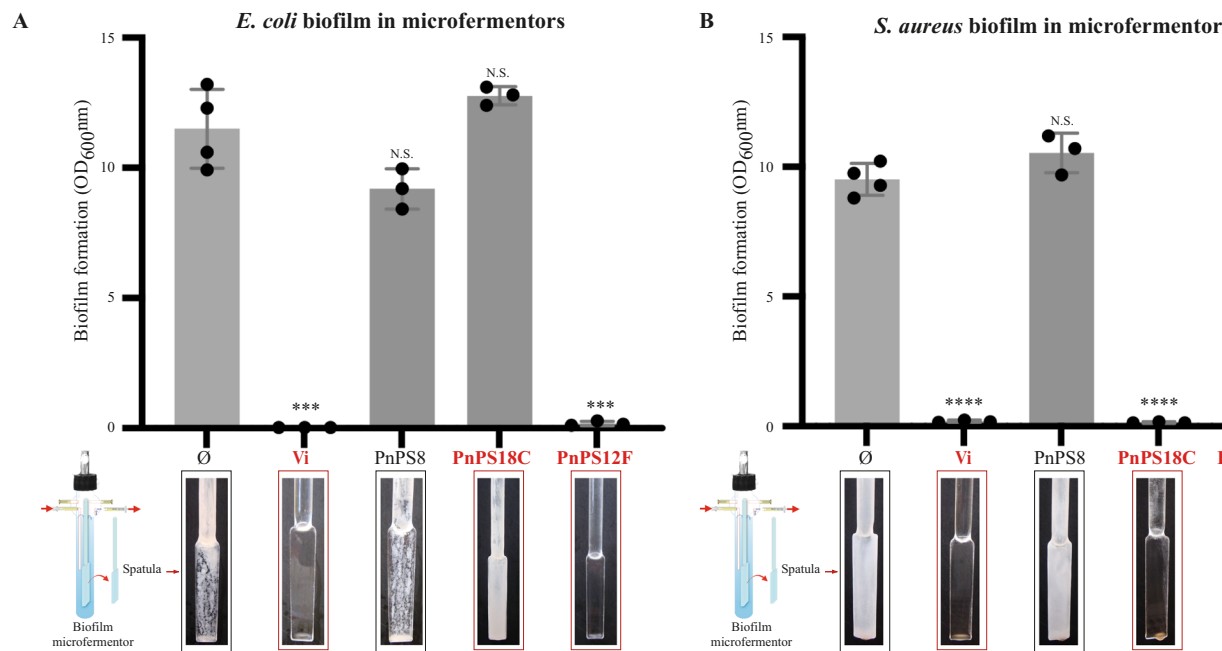

**Fig. 3 | Biofilm inhibitory effect of polysaccharides in continuous flow biofilm microfermentors. A** Quantification of biofilm formed in microfermentors by *E. coli* strain in M63B1glu medium, supplemented or not with 100 μg/ml Vi, PnPS8, PnPS18C or PnPS12F, and pictures of representative corresponding spatulas after 24 h of continuous growth. Each experiment corresponds to $n = 3$ biologically independent experiments, except for the control ($n = 4$). Source data are provided as a Source Data file. Statistical analyzes correspond to two-tailed unpaired *t*-test with Welch correction. Error bars represent SD. **B** Quantification of biofilm formed

in microfermentors by *S. aureus* strain in TSB medium, supplemented or not with 100 μg/ml Vi, PnPS8, PnPS18C or PnPS12F, and pictures of representative corresponding spatula after 24 h of continuous growth. Each experiment corresponds to $n = 3$ biologically independent experiments, except for the control ($n = 4$). Source data are provided as a Source Data file. Statistical analyzes correspond to two-tailed unpaired *t*-test with Welch correction. ***$p < 0.001$; ****$p < 0.0001$. Error bars represent SD.

---

including a biofilm-forming *E. coli* K-12 MG1655 carrying an F conjugative plasmid, *Enterobacter cloacae* 1092, *K. pneumoniae* Kp21, *S. aureus* 15981, and *S. epidermidis* O47 (Supplementary Fig. 4).

## Polysaccharide conformation and high intrinsic viscosity are predictive indicators of antibiofilm activity

To identify the structural and biophysical properties that potentially correlate with polysaccharide antibiofilm activity, we used HPSEC to determine the molecular weight ($M_w$) and the intrinsic viscosity ($[\eta]$) of the 8 identified active polysaccharides (G2cps, Vi, MenA, MenC, PnPS3, PRP, PnPS12F, PnPS18C) and 21 inactive polysaccharides (Table 1). Intrinsic viscosity $[\eta]$ of a given polysaccharide reflects its contribution to the viscosity $\eta$ of the whole solution, i.e. $[\eta] = (\eta - \eta_o)/(\eta_o \, \phi)$, where $\eta_o$ is the solution viscosity in the absence of polysaccharide and $\phi$ stands for the volume fraction of polysaccharides in solution. Accordingly, $[\eta]$ depends on the conformation adopted by the polysaccharides in the solution. This conformation is itself mediated by several physicochemical parameters, including the electrostatic charges carried by the polysaccharide and the charge distribution within the macromolecular body. The combination of $M_w$ and $[\eta]$ parameters are indicative of the volume (per mass unit) occupied by the polysaccharides in the solution. This analysis revealed a remarkable correlation between intrinsic viscosity and broad-spectrum antibiofilm activity. Indeed, all inactive macromolecules systematically display the lowest values of $[\eta]$, whereas active polysaccharides were characterized by a high (> 7 dl/g) intrinsic viscosity (Supplementary Fig. 5). Moreover, molecular weight $M_w$ and intrinsic viscosity $[\eta]$ of polysaccharides with intermediate narrow-spectrum activity (PnPS18C and PnPS12F) cover range of values measured for both broad-spectrum active and non-active macromolecules. To determine whether high intrinsic viscosity could be indicative of potential antibiofilm activity, we screened additional purified bacterial

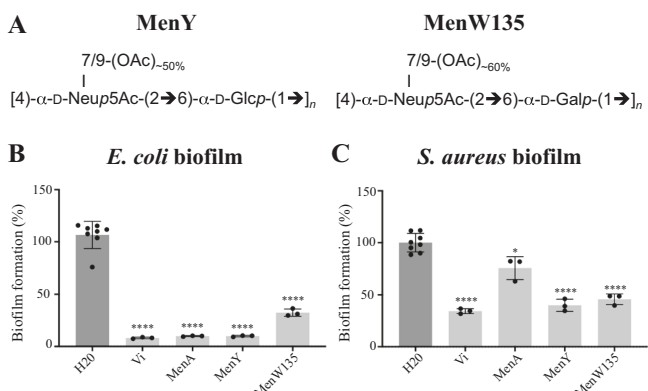

**Fig. 4 | MenY and MenW135 antibiofilm activity. A** Chemical composition and acidic/hydrophobic group ratio of the *N. meningitidis* polysaccharides MenY and MenW135. Antibiofilm activity against *E. coli* (**B**) and *S. aureus* (**C**) with MenY and MenW135 in comparison with previously identified active macromolecules Vi and MenA. Biofilm inhibition tests were performed in the presence of 50 μg/mL of polysaccharide. Distilled water was used as a negative control. Each experiment corresponds to $n = 3$ biologically independent experiments, except for the control ($n = 8$). Source data are provided as a Source Data file. Statistical analyzes correspond to two-tailed unpaired *t*-test with Welch correction. **$p < 0.01$; ***$p < 0.001$; ****$p < 0.0001$. Error bars represent SD.

capsular polysaccharides of our collection and we identified two such non-biocidal polysaccharides, MenY and MenW135 (Supplementary Fig. 3), presenting high intrinsic viscosity (Table 1 and Supplementary Fig. 5). Although MenY and MenW135 differ in their primary composition from Vi, MenA, MenC, and G2cps, they both exhibited similar

**Table 1 | Origin and relevant physicochemical properties of the polysaccharides used in this work**

| Bacterial origin | Name | *Mw* [a] (kDa) | Intrinsic viscosity[a] [η] (dl/g) | Charge index *i*[b] |
|---|---|---|---|---|
| *Escherichia coli* CFT073 | **G2cps** | 780 | 5 | 0.5 |
| *Salmonella* Typhi | **Vi** | 280 | 10 | 1 |
| *Haemophilus influenzae* serotype b | **PRP** | 550 | 10 | 0.5 |
| *Neisseria meningitidis* serogroup A | **MenA** | 200 | 10 | 1 |
| *Neisseria meningitidis* serogroup C | **MenC** | 200 | 10 | 1 |
| *Neisseria meningitidis* serogroup Y | **MenY** | 600 | 10 | 0.5 |
| *Neisseria meningitidis* serogroup W135 | **MenW135** | 500 | 12 | 0.5 |
| *Streptococcus pneumoniae* serotype 12 F | PnPS12F | 1000 | 10 | 0.16 |
| *Streptococcus pneumoniae* serotype 18 C | PnPS18C | 300 | 3 | 0.2 |
| *Streptococcus pneumoniae* serotype 1 | PnPS1 | 500 | 4 | 0.33 |
| *Streptococcus pneumoniae* serotype 3 | **PnPS3** | 700 | 7 | 0.5 |
| *Streptococcus pneumoniae* serotype 9 V | PnPS9V | 500 | 2 | 0.2 |
| *Streptococcus pneumoniae* serotype 8 | PnPS8 | 300 | 6 | 0.25 |
| *Streptococcus pneumoniae* serotype 19 A | PnPS19A | 250 | 1 | 0.33 |
| *Streptococcus pneumoniae* serotype 7 F | PnPS7F | 1000 | 1 | 0 |
| *Streptococcus pneumoniae* serotype 22 F | PnPS22F | 800 | 2 | 0.2 |
| *Streptococcus pneumoniae* serotype 14 | PnPS14 | 500 | 1 | 0 |
| *Streptococcus pneumoniae* serotype 2 | PnPS2 | 1300 | 5 | 0.17 |
| *Streptococcus pneumoniae* serotype 4 | PnPS4 | 400 | 4 | 0.25 |
| *Streptococcus pneumoniae* serotype 5 | PnPS5 | 200 | 3 | 0.2 |
| *Streptococcus pneumoniae* serotype 6B | PnPS6B | 1150 | 3 | 0.25 |
| *Streptococcus pneumoniae* serotype 9 N | PnPS9N | 700 | 2 | 0.2 |
| *Streptococcus pneumoniae* serotype 10A | PnPS10A | 550 | 2 | 0.14 |
| *Streptococcus pneumoniae* serotype 11 A | PnPS11A | 1000 | 3 | 0.25 |
| *Streptococcus pneumoniae* serotype 15B | PnPS15B | 800 | 2 | 0.2 |
| *Streptococcus pneumoniae* serotype 17 F | PnPS17F | 900 | 3 | 0.14 |
| *Streptococcus pneumoniae* serotype 19 F | PnPS19F | 650 | 3 | 0.33 |
| *Streptococcus pneumoniae* serotype 20 | PnPS20 | 600 | 2 | 0.17 |
| *Streptococcus pneumoniae* serotype 23 F | PnPS23F | 1200 | 5 | 0.25 |
| *Streptococcus pneumoniae* serotype 33 F | PnPS33F | 1000 | 2 | 0 |
| noncapsulated *Streptococcus pneumoniae* strain CSR SCS2 | CWPS | Nd | Nd | Nd |

Indicated numbers correspond to mean values.
[a]The relative standard deviation of *Mw* and [η] determination is <10%.
In bold: active broad spectrum antibiofilm polysaccharides. Underlined: active narrow spectrum antibiofilm polysaccharides.
[b]The charge index *i* corresponds to the ratio of the number of acidic group to the number of monosaccharide residue in a repeating unit.
*Nd*, Not determined.

broad-spectrum antibiofilm activity (Fig. 4 and Supplementary Fig. 6). These results indicated that specific polysaccharide conformation, reflected by a high intrinsic viscosity[29], could be a determinant of antibiofilm activity.

**Active antibiofilm polysaccharides are highly negatively charged macromolecules**
To further identify the molecular bases of the biophysical properties displayed by active polysaccharides, we first compared the composition of the broad-spectrum polysaccharides (Vi, MenA, MenW135, MenY, MenC, G2cps, PRP, PnPS3) but could not identify a chemical feature common to all active molecules. Alternatively, we tested whether increased surface hydrophilicity, previously associated with G2cps polysaccharide properties[7,9], could correlate with the anti-biofilm activity of the newly identified active polysaccharides. However, the determination of the surface contact angle of a drop of water on hydrophilic glass and hydrophobic plastic surfaces treated with either distilled deionized water or active and inactive polysaccharides did not reveal any correlation between surface hydrophilicity/hydro-phobicity balance and antibiofilm activity (Supplementary Fig. 7). As

additional support of this finding, the characterization of polysaccharide-coated surfaces by Atomic Force Microscopy (AFM) operated in force spectroscopy mode using $CH_3$-functionalized nanometric tips did not reveal any correlation between antibiofilm activity, structural surface patterns of adsorbed polysaccharides and surface hydrophobicity/hydrophilicity (Supplementary Fig. 8). We also evaluated the macromolecular charge, defined by a charge index *i* corresponding to the ratio between the number of acidic groups (carried by uronic and neuraminic acid groups or by phosphate groups) and the number of monosaccharide residues in a repeating unit (Table 1). The determination of this charge index showed that all active polysaccharides (Vi, MenA, MenW135, MenY, MenC, G2cps, PRP, PnPS3) correspond to highest *i*-values with *i* = 0.5 or 1. By contrast, macromolecules without antibiofilm activity and with narrow-spectrum activity could not be differentiated based on their respective *i*-values. For instance, PnPS18C (narrow activity), shares similar value of *i* (=0.2) with PnPS9V and PnPS22F (no activity). These results suggested that the density of negative charges carried by the poly-saccharides could be a key determinant of the biophysical properties associated with broad-spectrum antibiofilm activity.

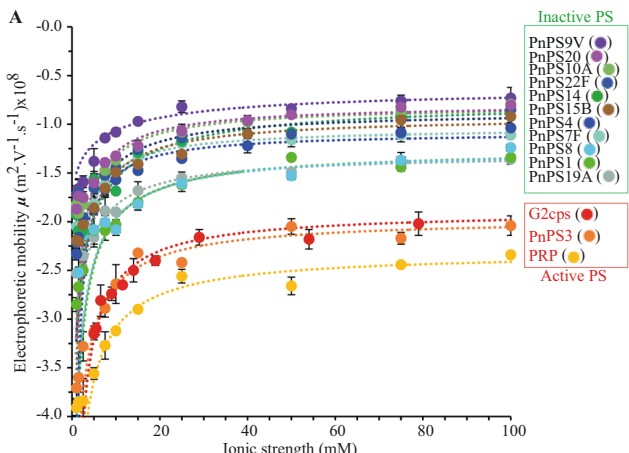

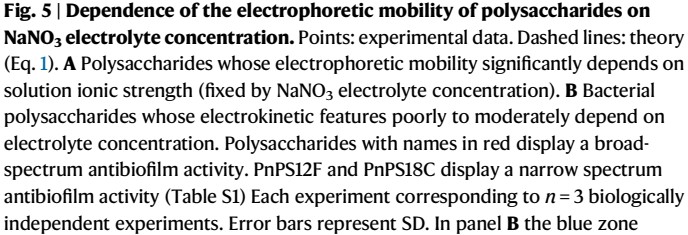

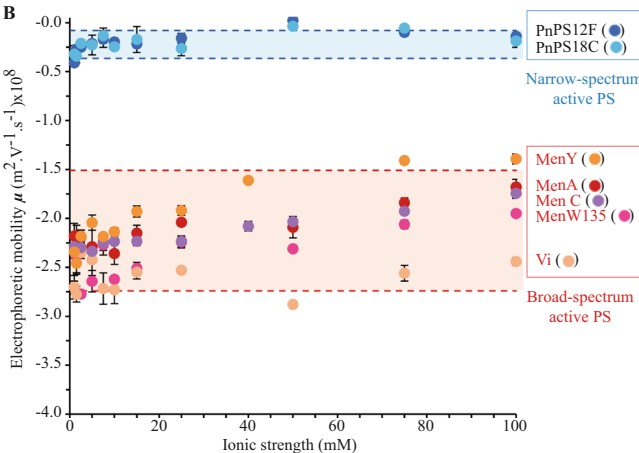

**Fig. 5 | Dependence of the electrophoretic mobility of polysaccharides on NaNO₃ electrolyte concentration.** Points: experimental data. Dashed lines: theory (Eq. 1). **A** Polysaccharides whose electrophoretic mobility significantly depends on solution ionic strength (fixed by NaNO₃ electrolyte concentration). **B** Bacterial polysaccharides whose electrokinetic features poorly to moderately depend on electrolyte concentration. Polysaccharides with names in red display a broad-spectrum antibiofilm activity. PnPS12F and PnPS18C display a narrow spectrum antibiofilm activity (Table S1) Each experiment corresponding to $n = 3$ biologically independent experiments. Error bars represent SD. In panel **B** the blue zone

delimited by blue dotted lines brackets the quasi-electrolyte concentration-independent electrophoretic mobility values measured for the polysaccharides with narrow-spectrum activity. The red zone delimited by red dotted lines brackets the quasi-electrolyte concentration-independent electrophoretic mobility values measured for the polysaccharides with broad spectrum activity. Each data point corresponds to $n = 3$ biologically independent experiments. Source data are provided as a Source Data file. Error bars represent SD. If not visible, error bars are hidden within their corresponding data point.

## Active and inactive antibiofilm polysaccharides display distinct electrokinetic patterns

Previous reports documented the link between structure, surface charge organization, and electrokinetic properties of macromolecules[30–32] as derived from electrophoresis. Bacterial polysaccharides are paradigms of 'soft' macromolecules, i.e. polyelectrolytic assemblies defined by a 3-dimensional charge distribution, permeable to ions from the background electrolyte solution and to the electroosmotic flow developed under electrophoresis measuring conditions[30–32].

To determine whether the electrokinetic properties of polysaccharides (which include both charge density and flow permeability features) are connected or not to their antibiofilm activity, we performed blind measurements of the electrophoretic mobility ($\mu$) as a function of NaNO₃ electrolyte concentration in solution (1 mM to 100 mM range) for all 8 broad spectrum (Vi, MenA, MenC, MenY, MenW135, G2cps, PRP, PnPS3) and 2 narrow spectrum (PnPS18C and PnPS12F) polysaccharides.

The electrophoretic mobility of these active polysaccharides was then compared to those of 11 inactive polysaccharides (PnPS1, PnPS4, PnPS9V, PnPS8, PnPS19A, PnPS20, PNPS10A, PnPS15B, PnPS7F, PnPS22F, and PnPS14) (Fig. 5). All tested polysaccharides displayed the characteristic electrophoretic signature expected for soft macromolecules, with an electrophoretic mobility tending to a constant, non-zero mobility plateau value (noted below as $\mu^*$) at sufficiently large salt concentrations (>30 mM). This property is the direct consequence of the penetration of the electroosmotic flow within the charged polysaccharide globular structure[33]. Further qualitative inspection of the sets of electrokinetic data collected for the polysaccharides of interest revealed two main electrokinetic patterns. The first one corresponded to the 11 tested inactive macromolecules whose electrophoretic mobility systematically tends to a value of $\mu^*$ satisfying $0.5 < |\mu^*| < 1.5 \times 10^{-8}\, \mathrm{m^2\,V^{-1}\,s^{-1}}$. For these macromolecules, the absolute value of the electrophoretic mobility decreases with increasing electrolyte concentration as a result of screening of the polysaccharide charges by the electrolyte ions. This feature is also shared by the active macromolecules (PnPS3, PRP and G2cps) with the noticeable difference that their asymptotic mobility value $|\mu^*|$ is significantly larger

with $\mu^*$ now satisfying the inequality $|\mu^*| > 2 \times 10^{-8}\, \mathrm{m^2\,V^{-1}\,s^{-1}}$ (Fig. 5A). The second observed electrokinetic pattern applies to macromolecules with narrow (PnPS18C and PnPS12F) or broad-spectrum (Vi, MenA, MenC, MenY and MenW135) activities for which the electrophoretic mobility $\mu$ moderately or poorly depends on background electrolyte concentration. Strikingly, active macromolecules with narrow-spectrum antibiofilm activity are defined by electrophoretic mobilities ($|\mu^*| < 0.5 \times 10^{-8}\, \mathrm{m^2\,V^{-1}\,s^{-1}}$) that are much lower in magnitude compared to those measured for broad-spectrum antibiofilm polysaccharides (Vi, MenA, MenC, MenY, and MenW135, $|\mu^*| > 1.5 \times 10^{-8}\, \mathrm{m^2\,V^{-1}\,s^{-1}}$) (Fig. 5B). These results demonstrated clearly that active capsular polysaccharides are characterized by a specific electrokinetic signature.

## The electrokinetic signature of antibiofilm polysaccharides is associated to their high flow permeability and high density of electrostatic charges

To further explore the electrokinetic properties of active capsular polysaccharides we interpreted quantitatively the dependence of the electrophoretic mobility ($\mu$) of the tested macromolecules on NaNO₃ electrolyte concentration with soft surface electrokinetic theory. This theory was developed by Ohshima for the electrophoresis of soft colloids[33] in the Hermans–Fujita's limit that is applicable to soft polyelectrolytes[34] (Eq. 1 in Materials and Method section). To that end, the required hydrodynamic radii of the macromolecules of interest were determined by Dynamic Light Scattering (DLS) after conversion of the measured diffusion coefficients by means of Stokes–Einstein equation (See Methods and Supplementary Table 2). This quantitative analysis highlighted a proper theoretical reconstruction of the electrophoresis data measured for all tested macromolecules (Fig. 5) and, most importantly, a remarkable classification of their activity (Fig. 6) according to their electrostatic and flow permeability properties.

These properties are expressed by two key parameters retrieved from the theoretical fitting of electrophoretic data to Eq. 1, namely: $\rho_o$, which is the net density of negative charges carried by the macromolecule, and $\lambda_o$ the so-called softness parameter. $1/\lambda_o$ (also called the Brinkman length) corresponds to the extent of penetration of the electroosmotic flow within the macromolecule. $1/\lambda_o$ is intimately

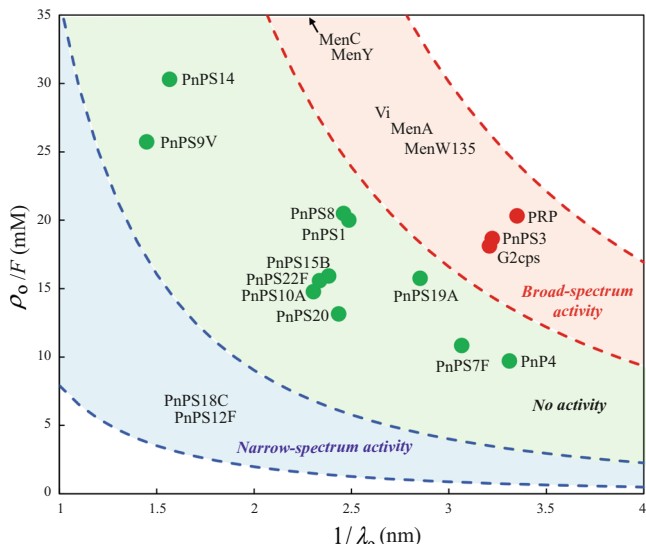

**Fig. 6 | Classification of the antibiofilm activity of the tested polysaccharides according to their charge density $\rho_o$ and flow penetration length scale $1/\lambda_o$.** The charge density $\rho_o$ (in C m$^{-3}$) is given here in the form of an equivalent concentration of anionic charges defined by $\rho_o/F$ (in mM) with $F$ the Faraday constant. The couple ($\rho_o,1/\lambda_o$) associated to each macromolecule is retrieved from the modeling of the dependence of measured electrophoretic mobility on the electrolyte concentration in solution according to Eq. 1. The dotted lines correspond to the set of solutions ($\rho_o,1/\lambda_o$) to the equation $\mu = \mu^*$ obtained for the macromolecules PnPS18C and PnPS12F and for the macromolecules Vi, MenW135 and MenA (see colored parts in Fig. 5B) whose mobility $\mu$ over the whole range of electrolyte concentrations does not significantly deviate from the mobility value $\mu^*$ measured at large electrolyte concentrations (100 mM). For these macromolecules, the poor dependence of $\mu$ on electrolyte concentration renders difficult any accurate evaluation of both $\rho_o$ and $1/\lambda_o$ as electrokinetic data fitting then reduces to solve one equation (Eq. 1) with two unknowns ($\rho_o$ and $1/\lambda_o$). The space of solutions ($\rho_o,1/\lambda_o$) associated with Vi, MenW135, MenA and PnPS18C, PnPS12F correspond to the red and blue zones, respectively. They are delimited by dotted lines that correspond to the ($\rho_o,1/\lambda_o$) solutions of the equation $\mu = \mu^*$ where $\mu^*$ identifies with the polysaccharide mobilities marked by the dotted lines represented in Fig. 5B. The error associated with values of $\rho_o$ and $1/\lambda_o$ (derived from Levenberg–Marquardt fitting of the data given in Fig. 5) is ±5%. This error is estimated from the standard deviations of the fitted experimental data provided in Fig. 5. Source data are provided as a Source Data file.

correlated to the polysaccharide structural compacity and by the degree of entanglement of its constituting chains. All macromolecules with narrow spectrum activity (PnPS18C and PnPS12F) are in the lower left zone within the diagram reporting the determined value of $\rho_o$ as a function of the corresponding $1/\lambda_o$ (Fig. 6). By contrast, polysaccharides with a broad-spectrum activity (e.g. Vi, MenA, MenC, MenY, MenW135, PRP, PnPS3, G2cps) combine high charge density and high Brinkman length scale, and they are thus found at the upper right region in the $\rho_o$-$1/\lambda_o$ representation (Fig. 6). All tested inactive macromolecules (PnPS1, PnPS4, PnPS9V, PnPS8, PnPS19A, PnPS20, PNPS10A, PnPS15B, PnPS7F, PnPS22F, and PnPS14) are positioned in a region intermediate between those of the broad-spectrum active and narrow-spectrum active macromolecules. Taken together, our results indicate that despite the lack of a specific molecular motif associated with antibiofilm properties, the combination of a loose structure (i.e. a large permeability to flow due to a large number of intraparticle voids) and a high density of carried electrostatic charges is critical for polysaccharides to exhibit antibiofilm activity.

## Discussion

The inhibition of bacterial adhesion using surface-active compounds is viewed as a promising approach to prevent the key initial steps of bacterial biofilm formation. In this study, we identified a total of 9 new

non-biocidal antibiofilm macromolecules among a collection of 31 different high molecular-weight bacterial capsular polysaccharides of known structure and composition. This allowed us to compare their chemical, structural and electrokinetic properties to identify key molecular and biophysical determinants discriminating active polysaccharides from inactive ones.

The identified active polysaccharides are composed of a high number of repeating oligosaccharides with different composition and structure. This suggests that, beyond the specific molecular composition of the active polysaccharide, the determinant of their antibiofilm activity could rather derive from supramolecular features. Consistently, we showed that the size integrity of the G2cps polysaccharide is critical to maintain its antibiofilm activity. We also demonstrated that a high intrinsic viscosity is a property shared by broad spectrum active macromolecules, which displayed the highest charge index $i$ among all tested polysaccharides tested. The hydrodynamic radii measured for all tested macromolecules was shown to cover a relatively narrow range, between ca. 15 and 40 nm in hydrodynamic diameter (Supplementary Table 2). Such small variations in particle size cannot account for the well-differentiated range of intrinsic viscosity measured for non-active and active macromolecules (1 to 6 and 7 to 12 dl/g, respectively, see Supplementary Fig. 5), especially so as both active and non-active polysaccharides display identical molecular weight ranges (Supplementary Fig. 5). In contrast, the high intrinsic viscosities measured for active macromolecules correlate with their charge index $i$, indicative of a high charge density. This correlation could be due to the fact that viscosimetric properties of particle dispersions depend on particle electrostatic characteristics that govern the extent of so-called primary and secondary electroviscous effects[35,36]. These effects are a direct consequence of the presence of charged electric double layers (EDL) at the macromolecule/solution interface and of the ensuing particle-particle and particle-fluid electrohydrodynamic interactions: the larger the density of particle charges, the more significant electro-viscous effects become[35,36]. As a consequence, the viscosity of a solution containing macromolecules of similar size can significantly differ according to the density of their carried electrostatic charges[35,36]. A large flow penetration within the particles body (as revealed by a high Brinkman length, $1/\lambda_o$, see Fig. 6) is consistent with the existence of a relatively loose structure adopted by the macromolecules and a reduced frictional force they exert on electro-osmotic flow during electrophoresis, with a resulting significant electrophoretic mobility. Altogether, the identified properties of loose macromolecular structure combined with a high charge density and a related high intrinsic viscosity correlate with antibiofilm activity.

Analysis of the electrokinetic data revealed very distinct electrokinetic patterns associated with a broad and narrow spectrum of antibiofilm activities, whereas inactive macromolecules exhibited an intermediate electrokinetic behavior (Fig. 6). This suggests that the high and low magnitudes of the polysaccharide charges could determine their broad and narrow activity, respectively, since a charge density with magnitude lying in between these two extremes led to a loss of activity. However, we previously showed that, despite its high negative charge, G2cps displayed a low affinity for cationic dyes[7], suggesting that its interaction with the surrounding biotic or abiotic environment is not only driven by electrostatics but may also involve remodeling of its surface properties. This could possibly include changes in surface hydration or steric repulsion, and subsequent limitation of bacterial adhesion[7]. This indicates that consideration of polysaccharide electrostatic properties alone is not enough to account for their antibiofilm activity. Consistently, analysis of our electrokinetic data sets suggests indeed that the very organization of the polymer chains and the resulting flow permeability properties of the polysaccharidic macromolecules (as qualitatively indicated by the magnitude of their Brinkman length) play an important role in defining their

antibiofilm activity. Considering that only charges located in the peripheral region of the macromolecules are probed by electrophoresis (35), the location of these charges and their degree of exposition to the outer solution are likely important factors underlying antibiofilm activity.

Then, how does polysaccharide molecular composition correlate with antibiofilm activity? Most active macromolecules display a high density of negative charges that could contribute to the electrostatic repulsion of negatively charged bacteria. However, bacterial surface structures such as pili, fimbriae, lipopolysaccharides, or even polycationic exopolysaccharides are known to overcome these repulsive forces and promote bacterial adhesion and aggregation[37–39]. We, therefore, hypothesize that the directionality of the interactions on surfaces[40] could be a critical determinant of broad-spectrum antibiofilm polysaccharide activity. The proper exposition of electrostatic charges could optimize the repulsion effects in the antibiofilm macromolecules identified in this study. In addition, their high intrinsic viscosity and relative loose structure could mechanically modify the local conditions of adhesion and alter the perception of the surface by bacteria, thereby minimizing their adhesion in the presence of the identified polysaccharides[41]. We, therefore, propose that the active antibiofilm polysaccharides identified in our study could have multipronged activity, with both short and long-range modifications of the surface-bacteria or bacteria-bacteria interactions, in particular via an alteration of the local adhesion conditions prevailing on the adhesion surface. The distinction between broad, narrow, or lack of activity would therefore depend on the specific combination of bio-physical-chemical properties displayed by each macromolecule.

By providing a better definition of the chemical and structural basis of the broad-spectrum antibiofilm activity displayed by capsular polysaccharides, our study offers the possibility to identify new surface-active polysaccharides on the basis of their biophysical properties but also to design and engineer macromolecules mimicking antibiofilm polysaccharide activity through total or partial synthesis. These molecules could be used as non-biocidal biofilm control strategies in prophylactic treatment against the initial adhesion of biofilm-forming pathogens developing on medical and industrial materials.

## Methods
### Bacterial strains and growth conditions
The bacterial strains used in this study are listed in Supplementary Table 3. Gram- bacteria were grown in 0.4% glucose-M63B1 minimal medium (M63B1) or in lysogeny broth (LB) medium at 37 °C, with appropriate antibiotics when required. Gram+ strains were cultured in tryptic soy broth (TSB) supplemented with 0.25% glucose. The potential biocidal effect of antibiofilm polysaccharides was evaluated from growth curve determination in the presence of 100 µg/ml of purified polysaccharide.

### Bacterial polysaccharides
Capsular polysaccharides used in this study were obtained from Sanofi, Marcy l'Etoile France, and Swiftwater, USA. *E. coli* polysaccharides were produced at the Institut Pasteur, Paris. Teichoic acid polysaccharide (cell wall PS, CWPS) from *Streptococcus pneumoniae* strain CSR SCS2 was from Statens Serum Institut (Ref 3459).

### Biofilm formation and inhibition assays
**Static tests: microtiter plate crystal violet biofilm assay.** Overnight cultures were adjusted to an $OD_{600nm}$ of 0.02 before inoculating 50 µl into 96-well polyvinyl chloride (PVC) plates (Falcon; Becton Dickinson Labware, Oxnard, CA) and added at a 1:1 ratio to 50 µl of filter-sterilized purified polysaccharide at the desired concentration: i) 50 µg/ml for standardized antibiofilm activity experiments; ii) 3.125–100 µg/ml to test the level of activity of the macromolecules. Biofilms were left to grow for 16 h at 37 °C. The supernatant was then removed from each

well, washed three times with water and biofilm was stained with 1% crystal violet solution for 15 min. After removal of the crystal violet solution, biofilms were washed three times with water. For quantification of biofilm formation, dried stained biofilms were resuspended in 30% acetic acid and absorbance was measured at 585 nm using an infinite Tecan infinite M200 PRO plate reader. Biofilm formation with different strains is represented in values normalized to average biofilm formation with the control strains.

**Dynamic tests: continuous flow biofilm microfermentor assay.** Pyrex biofilm microfermentors containing a removable Pyrex glass spatula (V.S.N., Paris, France, ref. 028703022601) were used to grow biofilms under continuous flow conditions, as described in[42] (see also https://research.pasteur.fr/en/tool/biofilm-microfermenters/). The medium flow was adjusted to 60 ml/h with internal bubbling agitation using filter-sterilized compressed air. This flow rate minimizes planktonic growth over biofilm development by constant dilution of non-biofilm bacteria. Inoculation was performed by dipping the glass spatula for 5 min in $OD_{600} = 1$ M63B1 0.4% glucose minimal medium (M63B1glu) for *E. coli* strain and in tryptic soy broth 0,25% glucose (TSBglu) for *S. aureus* strain, supplemented or not with 100 µg/ml purified polysaccharide Vi, PnPS8, PnPS18C, PnPS12F. After bacteria were allowed to attach to the spatulas, they were then reintroduced into the microfermentor filled with the appropriate media supplemented also with the same final concentration of tested polysaccharide. After 24 h of continuous culture, pictures of the glass spatula were taken and biofilm biomass was estimated by determining the $OD_{600nm}$ of the biofilm resuspended from the internal microfermentor glass slide.

### Test of non-biocidal activity
Growth curves of *S. aureus* and *E. coli* exposed to 100 µg/ml were measured in the presence of each indicated tested macromolecule (Supplementary Fig. 3). The bacterial strains were inoculated in microtiter plates at $OD_{600nm}$ of 0.05 and let to grow in TECAN plate reader (Infinite 200 Pro) with a 2 mm orbital shaking amplitude for 24 h at 37 °C. Bacterial growth was determined by the TECAN plate reader.

### Polysaccharide purification
G2cps polysaccharide was obtained from 24 h cultures as described in[7]. Briefly, CFT073 *E. coli* strains were grown in M63B1 0.4% glucose for 24 h at 37 °C. After cold (4 °C) centrifugation at 3800 g for 10 min, supernatants were filtered through a 0.22 µm filter. Polysaccharides were precipitated from cell-free supernatant using ice-cold ethanol (3:1 ratio ethanol/supernatant) followed by centrifugation for 2 h at 5000 g and 4 °C. Precipitated polysaccharides were resuspended and dialyzed against deionized water (Slide-A-Lyzer Dialysis Cassettes, 10K MWCO, Pierce, Rockford, IL). Total polysaccharide concentrations were measured by Dubois colorimetric assay[43]. The polysaccharides were finally separated from residual lipopolysaccharides by gel filtration on Sepharose 6BCL column (GE Healthcare Life Sciences) in 1% sodium deoxycholate[44] and dialyzed against deionized water. Polysaccharide concentration was determined using High-Performance Anion-Exchange Chromatography with Pulsed Amperometry Detection (HPAEC-PAD) (Thermo Fischer Scientific, Dionex, Sunnyval, CA) as previously described[45].

### Polysaccharide structure analysis by high performance anion exchange chromatography with pulsed amperometric detection (HPAEC-PAD)
Polysaccharide G2cps was hydrolyzed with hydrofluoric acid (HF) (48% by mass) for 16 h at room temperature. HF was removed by drying under a stream of nitrogen at 40 °C. The sample was redissolved in water and then hydrolyzed with 2 M trifluoroacetic acid (TFA) for 2 h at

121 °C. TFA was removed by drying under a stream of nitrogen at 40 °C. The sample was redissolved in water prior to analysis. For the sake of comparison, both HF-alone and TFA-alone hydrolysis were also performed on polysaccharide G2cps. Pulsed amperometric detection was used incorporating a quadruple-potential waveform. Data were collected and analyzed on computers equipped with Dionex Chromeleon software (Dionex). Commercial monosaccharides were used as standards.

## Nuclear magnetic resonance (NMR) spectroscopy

Approximately 5 mg of polysaccharide was lyophilized once, dissolved in 700 µl of deuterated water and 550 µl were introduced in a 5 mm tube. NMR spectra were collected on a Bruker Avance 500 MHz spectrometer running Topspin 2.1 software at an indicated probe temperature of 20 °C. Spectra were measured for solutions in $D_2O$ with 0.01% DSS as an internal standard for proton ($^1H$) NMR (δ 0.00 ppm). Phosphoric acid (2%) was used as an external standard for phosphorus ($^{31}P$) NMR (δ 0.00 ppm) and TSP as an external standard for carbone ($^{13}C$) NMR (δ 0.00 ppm). The pulse programs used were those in the Bruker library except for the correlation spectroscopy (COSY) and total correlation spectroscopy (TOCSY) diffusion experiments. The diffusion delay was 80 ms for these two-dimensional experiments and the mixing time for the TOCSY was 100 ms. The two-dimensional $^1H$-$^{31}P$ experiment was acquired using a $J_{H-P}$ coupling value of 7 Hz. The $^1H$-$^{13}C$ heteronuclear single quantum coherence (HSQC) spectra were acquired using a $^nJ_{H-C}$ coupling value of 145 Hz and a long range $^nJ_{H-C}$ coupling value of 5 Hz for heteronuclear multiple bond correlation (HMBC).

## High performance size exclusion chromatography methods with triple detection: $Mw$ and $[\eta]$ determinations

Polysaccharides were analyzed at a concentration between 600 to 1000 µg/ml in water. Analyzes were performed using a TDA301 Viscotek HPSEC system consisting of an automated sampler with a HPLC pump equipped with an injection loop of 100 µl, an on-line degasser, a viscometer, a refractive index (RI), and a right-angle laser light scattering detector (RALLS). The HPSEC analyzes of polysaccharide were performed using TSK G4000 PWXL (0.7 × 30 cm) and TSK G6000 PWXL analytical columns (0.7 × 30 cm) connected in series at a flow rate of 0.5 ml/min, at 30 °C. Tris buffer 2.5 mM, pH 7.5, or phosphate buffer 200 mM pH 6.9 was used as the mobile phase. The columns were kept at a constant temperature of 30 °C. Omnisec software (Malvern) was used for data collection and data processing. This triple detection consisting of online RI, RALLS, and viscometric detector was used to determine the molecular weight ($Mw$) and the intrinsic viscosity ($[\eta]$) of polysaccharides.

## Size reduction of G2cps polysaccharide

A total of 7.5 mg of polysaccharide were hydrolyzed in a solution with ascorbic acid, copper, and iron sulfate at 30 °C. Hydrolysis kinetics was followed by HPSEC analysis for 15 min, 1 h, and 24 h.

## Contact angle measurements

Glass coverslip slides (24×50mm Menzel-Glaser Thermo Scientific ref17234914) or 17 × 28 mm polyester plastic coverslip slides (Thermo scientific ref AB-0578) were treated with 80 µl of distilled deionized water or with 100 µg/ml solution of purified active and inactive polysaccharides. Slides were dried under a laminar flow hood and 2.5 µl drops of water were deposited using a Kruss DSA4 Drop Shape Analyzer on the surface of coated and uncoated slides, and contact angle was then measured 3 times.

## Atomic force microscopy measurements

Microscopy glass slides were treated with 100 µg/ml solution of purified active and inactive polysaccharides for 10 min and rinsed with ultrapure water. AFM measurements were performed in ultrapure water, using a Fastscan Dimension Icon with Nanoscope V controller (Bruker). Images were obtained in Peak Force tapping mode, using gold-coated silicon nitride tips (NPG, Bruker), with a maximum applied force of 500 pN, a scan rate of 1 Hz, a peak force amplitude of 300 nm, and a peak force frequency of 2 kHz. For quantifying coated-surface hydrophobicity by means of chemical force microscopy, hydrophobic tips were prepared by immersing gold-coated silicon nitride tips (NPG, Bruker) for 12 h in 1 mM solution of dodecanethiol (Sigma) in ethanol, rinsed with ethanol and dried with $N_2$. Spatial mappings were obtained by recording 32 × 32 approach-retract force-distance curves on 5 µm × 5 µm areas, with a maximum applied force of 500 pN, and an approach and retraction speed of 500 nm/s.

## Electrokinetic measurements

1 g/L stock suspensions of macromolecules were first prepared in ultrapure water from corresponding frozen powders and, after 48 h, suspensions were diluted at 250 mg/L and adjusted to pH 5. For each macromolecule tested, batches were then prepared in order to obtain a series of suspensions with $NaNO_3$ background electrolyte concentration in the range 1 to 100 mM. Final macromolecule concentration and pH conditions were adopted to optimize the measured electrophoretic response after testing a range of particle concentrations from 50 to 250 mg/L and four pH conditions (3, 5, 7, and 9). All dispersions, stock, and diluted suspensions were stored at 4 °C and were re-acclimatized at room temperature prior to measurements.

After 24 to 48 h of batches preparation, diffusion coefficients (converted into particle diameter using Stokes–Einstein equation) and electrophoretic mobilities of the bacterial polysaccharides were measured by Dynamic Light Scattering (DLS, Supplementary Table 2) and Phase Analysis Light Scattering (Malvern Instruments). Each reported data point in Fig. 5 corresponds to three measurements performed on 3 different aliquots of a given batch macromolecule dispersion.

## Quantitative assessment of polysaccharide electrophoretic mobility

Electrophoretic mobility of the various macromolecules of interest was measured as a function of $NaNO_3$ electrolyte concentration (Fig. 5) and was collated with the analytical theory developed by Ohshima for electrophoresis of soft colloids[33] in the Hermans-Fujita limit applicable to soft polyelectrolytes[34]. According to the latter, $\mu$ is defined by the expression

$$\mu = \frac{\rho_o}{\eta\lambda_o^2}\left\{1 + \frac{1}{3}\left(\frac{\lambda_o}{\kappa}\right)^2\left[1 + e^{-2\kappa b} - \frac{1 - e^{-2\kappa b}}{\kappa b}\right]\right.$$
$$\left. + \frac{1}{3}\left(\frac{\lambda_o}{\kappa}\right)^2 \frac{1 + 1/\kappa b}{(\lambda_o/\kappa)^2 - 1}\left[\left(\frac{\lambda_o}{\kappa}\right)\frac{1 + e^{-2\kappa b} - (1 - e^{-2\kappa b})/\kappa b}{(1 + e^{-2\lambda_o b})/(1 - e^{-2\lambda_o b}) - 1/\lambda_o b} - \left(1 - e^{-2\kappa b}\right)\right]\right\},$$

(1)

where $\rho_o$ represents the density of negative charges carried by the macromolecule, $\kappa$ is the reciprocal Debye layer thickness defined by $\kappa = [2F^2c_o/(\varepsilon RT)]^{1/2}$ with $R$ the gas constant, $T$ the absolute temperature, $F$ the Faraday number and $c_o$ the bulk concentration of a 1:1 electrolyte ($NaNO_3$ in this work), $\lambda_o$ is the so-called softness parameter with $1/\lambda_o$ corresponding to the characteristic penetration length of the electroosmotic flow within the macromolecule. The quantity $b$ in Eq. 1 is the radius of the polyelectrolyte macromolecule. The set of ($\rho_o$, $1/\lambda_o$) couples obtained for all polysaccharidic macromolecules are reported in Supplementary Table 2 and the required values of $b$ for data fitting to Eq. 1 were determined by Dynamic Light Scattering and are also given in Supplementary Table 2. The dependence of electrophoretic mobility data on electrolyte concentration was fitted using Levenberg-Marquardt method with Mathcad 13 software.

## Statistical analysis

Two-tailed unpaired *t-test* with Welch correction analyzes were performed using Prism 9.0 for Mac OS X (GraphPad Software). Each experiment was performed at least three times. All data are expressed as mean (±standard deviation, SD) in the figures. Differences were considered statistically significant for *p* values of <0.05; *\*p* < 0.05; *\*\*p* < 0.01; *\*\*\*p* < 0.001. *\*\*\*\*p* < 0.0001.

## Reporting summary

Further information on research design is available in the Nature Portfolio Reporting Summary linked to this article.

## Data availability

All data generated in this study and its supplementary Information are provided as a Supplementary Source Data file in the Supplementary Information. This Microsoft Excel file provides the source data displayed in the figures presented in the main text and in Supplementary Information. Source data are provided with this paper.

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

## Acknowledgements

We thank Olaya Rendueles for helpful discussions and Nadia Izadi and Laurence Mulard for critical reading of the manuscript. We are grateful to Heike Claus, Ulrich Vogel and Muhamed-kheir Taha, for generously providing us assistance with some of the strains used in this study. We also thank Sandrine Mariot from the Somacomic platform, Orsay, for her help in the measure of contact angles. This work was supported by a collaborative research grant Institut Pasteur and Sanofi, by grants from the French Government's Investissement d'Avenir program, Laboratoire d'Excellence "Integrative Biology of Emerging Infectious Diseases" (grant n°ANR-10-LABX-62-IBEID to JMG) and by the Fondation pour la Recherche Médicale (grant DEQ20180339185 to JMG). J. B-B was the recipient of a long-term post-doctoral fellowship from the Federation of European Biochemical Societies (FEBS) and by the European Union's Horizon 2020 research and innovation program under the Marie Skłodowska-Curie grant agreement No 842629. This work was partly carried out in the Pôle de Compétences Physico-Chimie de l'Environnement, LIEC laboratory UMR 7360 CNRS - Université de Lorraine.

## Author contributions

J.B.-B., N.M., P.T., J.F.L.D., and J.-M.G. designed the experiments. L.T., C.B., and B.R. contributed to the initial experiments. J.B.-B., J.T., M.B., A.B., C.C., Y.W., S.L., T.F., J.-M.G., and P.T. performed the experiments. J. B.-B., J.T., M.B., A.B., T.F., S.L., P.T., N.M., J.F.L.D., and J.-M.G. analyzed the data. J. B.-B., J.F.L.D., and J.-M.G. wrote the paper with significant contributions from C.B., J.T., T.F., A.B., P.T., B.R., and N.M.

## Competing interests

J.T., M.B., B.R., P.T., and N.M. are Sanofi employees and may hold shares or stock options in the company. The remaining authors declare no competing interests.

## Additional information

[1]Institut Pasteur Université Paris Cité, CNRS UMR 6047, Genetics of Biofilms laboratory, Paris F-75015, France. [2]Departamento de Genética, Facultad de Biología, Universidad de Sevilla, Apartado 1095, 41080 Sevilla, Spain. [3]Sanofi, Research & Development, Campus Mérieux, 1541 avenue Marcel Mérieux,, 69280 Marcy l'Etoile, France. [4]Université de Lorraine, CNRS, Laboratoire Interdisciplinaire des Environnements Continentaux (LIEC), F-54000 Nancy, France. [5]Institut Pasteur, Université Paris Cité, Inserm U1224, Brain-Immune Communication group, F-75015 Paris, France. [6]Institut Pasteur, Université Paris Cité, INRAE, USC2019, Fungal Biology and Pathogenicity laboratory, F-75015 Paris, France. ✉e-mail: Noelle.Mistretta@sanofi.com; jerome.duval@univ-lorraine.fr; jmghigo@pasteur.fr

