## [Peer Review File · Nature Communications]

Bacterial capsular polysaccharides with antibiofilm activity share common biophysical and electrokinetic propertiesREVIEWER COMMENTS

Reviewer #1 (Remarks to the Author):

The authors present a manuscript in which they highlight that bacterial capsular polysaccharides owe their antibiofilm potential to structural requirements as high intrinsic viscosity and specific electrokinetic signature. The results presented are very interesting and could reasonably be useful for the research and production of new antibiofilm agents. Also, the methodologies used seem appropriate as well as the observations that the authors have obtained and the way in which they have reworked them in the conclusions. The methods are well described and the equalizations used are also well motivated. On the basis of this premise, I believe that the work, however, needs an adequate review in some points that are ambiguous or even incorrect. In detail, I find the description of the number of capsules used in certain experiments very confusing in some points as several times I have not found a match between the number indicated in the text and the one actually indicated in the figures.

For example, on page 4 line 25, the authors plan to use 32 capsules purified by both Gram+ and Gram- bacterial strains. This number is never respected in any figure or table in the manuscript. On page 6 line 29, the authors declare that they screened 30 different high molecular-weight bacterial polysaccharides to detect properties similar to that G2cps, the latter used as a reference. Thus, the capsules should be 31 and this number is correctly reported in table S1. However, when the reader analyzes Figure 2, the capsules indicated in the graphs are 29 including G2cps. Authors should better specify or correct. Further clarifications are needed regarding the motivation for choosing 4 capsules and not 5 in the experiment described on page 10, line 7 and shown in figure S4.

Another apparent contradiction concerns the different number of capsules reported in the tables. Table 3 shows 17 capsules, table S2 shows 15. In the text, this difference should be better justified if supported by real reasons.

In figure S4 reference is made to several active polysaccharides. In reality these are 4 polysaccharides, the caption must be corrected.

Some typos are scattered throughout the text or figures.

Reviewer #2 (Remarks to the Author):

This study offers biophysical and elektrokinetic methods to identify new surface-active polysaccharides for antibiofilm activity.

The work is significant for antibiofilm activity. The antibiofilm activity is poorly studied and it is not clear if the authors studied a biofilm or planktonic bacteria from the method they used, see also comments below. Therefore it is suggested to add a better antibiofilm assay comparable as in the paper of Valle et al. 2006.

Additional antibiofilm methods are necessary to proof the antibiofilm activity of the isolated polysaccharides.

The question is how do you know you have a biofilm and not just a planktonic culture? Normally when you have a biofilm you first need to adhere bacteria to a surface from growth media or buffer, wash, add growth media and let the biofilm grow for the appropriate time. You need to add the quantification assay. T

Methods, in which medium where *S. aureus* and *S. epidermidis* cultured, which cultures get 0.25 and 0.50% glucose? Can you give a bit more details?

Specify cold centrifugation.

Check if all abbreviations are introduced.

The assay used for non-biocidal activity is lacking in the Methods, it is partly described in the caption, but growth media for example is lacking. Also add the agitation speed.

Can you use this on all surfaces? So far you tested it only on PVC?

Reviewer #3 (Remarks to the Author):

This manuscript is well written and organized. The authors have studied the biophysical properties of non-biocidal antibiofilm polysaccharides and the correlation between the properties and the antibiofilm activity in order to examine new approaches to engineering and screening of antibiofilm polysaccharides. The methods are generally appropriate to fulfill the overall aim of the study, but some conclusions drawn in the result section should be carefully revised to be more specific and take into consideration the study limitations and the sample size. Specific comments follow. The authors make a systematic contribution to the research literature in this area of investigation. Overall, this is a good-quality manuscript with interesting new highlights on the biophysical properties shared by polysaccharides with antibiofilm activity.

ABSTRACT:

Page 2, Line 11-13

“we screened a collection of 32 purified capsular polysaccharides and identified seven new compounds with non-biocidal activity against biofilms formed by *Escherichia coli* and/or *Staphylococcus aureus*.”

In the section page 4 line 25 and page 6 line 31, The compounds are described as previously known. Therefore, the statement “seven new compounds” as an “novel” should be revised or clarified to reflect that the antibiofilm activity is the new finding. Please, revise this statement throughout the paper.

“the polysaccharide mobility”. Mobility of all polysaccharides was assessed

Page 2, Line 17-18

“Based on these characteristics, we identified two additional antibiofilm capsular polysaccharides with high density of electrostatic charges and their permeability to fluid flow.”

distinct electrokinetic properties

Page 2, Line 21-23

“This characterization of a specific electrokinetic signature for polysaccharides displaying antibiofilm activity opens new perspectives to identify or engineer non-biocidal surface-active macromolecules to control biofilm formation in medical and industrial settings.”

INTRODUCTION

Page 3, Line 4

“negatively impact human activities”. What is meant by activities? Biofilm developing on medical or industrial surfaces affect human health.

Page 3, Line 12-14

“These approaches are limited by the rapid masking of the coated surfaces by bacterial and organic debris, and are associated with worrisome selection of resistance upon repeated contact with treated surfaces”. This sentence needs to be reworded. It is not clear if it is bacterial debris or actual bacteria and organic debris, the resistance is not specified as antibiotic resistance, and word masking doesn't seem to be used appropriately. Also, what does “upon repeated contact refers to?

Page 3, Line 22:

“to protect patient care equipment” is unclear as to what type of protection. Consider removing or clarifying. Suggestion: “to protect patient care equipment from pathogen colonization and...”

Page 3, Line 25

“bio-sourced strategies”. Do you mean biobased strategies?

Page 3, Line 25-27

“preventing and/or disrupting biofilms based on the use of quorum sensing inhibitors”. Suggested rewording the sentence as “preventing and/or disrupting biofilm formation using quorum sensing inhibitors”

RESULTS

Page 7, Line 17

statement title “Polysaccharide conformation and high intrinsic viscosity are predictive indicators of antibiofilm activity” is an overstatement in the result section and should be revised as it is “a potentially predictive indicator” at this point.

Page 7, Line 22, Page 10, Line 2, and other result sections:

“and inactive (PnPS1, PnPS8, PnPS19A, PnPS9V, PnPS7F, PnPS22F and PnPS14) polysaccharides by HPSEC”. “compared to those of seven inactive polysaccharides (PnPS1, PnPS9V, PnPS8, PnPS19A, PnPS7F, PnPS22F and PnPS14)”

Why and how these specific inactive polysaccharides were selected to compare their properties to active compounds since 32 Polysaccharides were screened and at least 23 were inactive.

Have all inactive compounds been tested for their intrinsic viscosity and for their electrokinetic behavior?

Page 8, line14

“These results indicated that 14 specific polysaccharide conformation, reflected by a high intrinsic viscosity (Yang 15 1961), is a determinant of antibiofilm activity.” This study showed that high intrinsic viscosity can be a “predictive indicator of antibiofilm activity”. However, because of the sample size and other properties evaluated, the concluding sentence(is a determinant of antibiofilm activity) of this paragraph appears to be an overstatement.

Reviewer #1 (Remarks for the Author):

The authors present a manuscript in which they highlight that bacterial capsular polysaccharides owe their antibiofilm potential to structural requirements as high intrinsic viscosity and specific electrokinetic signature. The results presented are very interesting and could reasonably be useful for the research and production of new antibiofilm agents. Also, the methodologies used seem appropriate as well as the observations that the authors have obtained and the way in which they have reworked them in the conclusions. The methods are well described and the equalizations used are also well motivated.

On the basis of this premise, I believe that the work, however, needs an adequate review in some points that are ambiguous or even incorrect.

1. In detail, I find the description of the number of capsules used in certain experiments very confusing in some points as several times I have not found a match between the number indicated in the text and the one actually indicated in the figures.

For example, on page 4 line 25, the authors plan to use 32 capsules purified by both Gram+ and Gram-bacterial strains. This number is never respected in any figure or table in the manuscript.

On page 6 line 29, the authors declare that they screened 30 different high molecular-weight bacterial polysaccharides to detect properties similar to that G2cps, the latter used as a reference. Thus, the capsules should be 31 and this number is correctly reported in table S1. However, when the reader analyzes Figure 2, the capsules indicated in the graphs are 29 including G2cps. Authors should better specify or correct.

We agree with the reviewer that the description of the number of capsules studied was confusing and we have now clarified this point in the revised version of the manuscript.

Briefly, in our initial screening of polysaccharides we investigated the activity of 28 capsules, in addition to the previously described Group2 capsule, hence the number of 30 capsular polysaccharide presented in the first result section was indeed incorrect and should instead read 29. This is now corrected in the corresponding result section.

Then, using high intrinsic viscosity as a predictive indicator of antibiofilm activity, we tested more capsules and we identified two additional broad-spectrum antibiofilm polysaccharides (MenY and MenW135), thus increasing our number of studied capsules to 31 (and not 32). This is now corrected in the abstract and in the discussion.

The structures of these 31 capsules are now correctly presented in Table S1. The 9 active molecules identified in the study (10 when including the previously known G2cps) are indicated in color (red and blue) in Table S1 and Table 1.

All remaining discrepancies noted by the reviewers regarding the number of studied capsules, have been corrected in the abstract, the introduction, the corresponding result sections, and in the discussion.

Further clarifications are needed regarding the motivation for choosing 4 capsules and not 5 in the experiment described on page 7, line 10 and shown in figure S4.

Among the 5 broad-spectrum capsular polysaccharides to be tested (Vi, MenA, MenC, PRP and PnPS3), we first did not include MenC because MenA and MenC displayed quite similar primary structure, molecular weight M_w and intrinsic viscosity. However, the reviewer has a good point and we now have addressed this issue by performing additional electrokinetic analyses so as to include more capsules

molecules, comprising all actives ones (including MenC, but also MenY and MenW135) along with 4 additional inactive polysaccharides (PnPS4, PnPS10, PnPS15B, PnPS20) selected for the diversity of their structure and composition.

This allowed us to provide a better representation of the electrokinetic properties of active and inactive capsular polysaccharides. This is now indicated in the revised version of the manuscript and in figures 5 and 6 – formerly 4 and 5- pasted below. Most remarkably, we fully confirm the classification we proposed in our original manuscript between broad-active, narrow active and inactive molecules on the basis of their distinct electrokinetic characteristics.

This is now indicated in the revised version of the text as follows

Page 10 Line 274-281: **Result section**

“... we performed blind measurements of the electrophoretic mobility (μ) as a function of NaNO₃ electrolyte concentration in solution (1mM to 100mM range) for all 8 broad spectrum (Vi, MenA, MenC, MenY, MenW135, G2cps, PRP, PnPS3) and 2 narrow spectrum (PnPS18C and PnPS12F) polysaccharides. The electrophoretic mobility of these active polysaccharides was then compared to those of 11 inactive polysaccharides (PnPS1, PnPS4, PnPS9V, PnPS8, PnPS19A, PnPS20, PnPS10A, PnPS15B, PnPS7F, PnPS22F and PnPS14) (Figure 5 and 6).”

Figure 5

Figure 6

Another apparent contradiction concerns the different number of capsules reported in the tables. Table 1 shows 17 capsules, table S2 shows 15. In the text, this difference should be better justified if supported by real reasons.

As indicated above, we apologize for the confusion but the initially submitted manuscript correctly indicated that Table 1 presented the intrinsic viscosity and molecular weight of 17 active and inactive capsules, whereas Table S2 only corresponds to the 15 capsules for which size and electrokinetic properties were obtained by Dynamic Light Scattering (DLS) and electrophoresis data subjected to modeling, respectively.

To clarify this aspect and further address other reviewer's concerns, the revised Table 1 now presents ALL 31 tested polysaccharides. Their molecular weight and intrinsic viscosity are now all systematically plotted in a more complete version of revised Figure S6 (formerly S5, pasted below). This new Figure again confirms the remarkable differentiation between active and inactive molecules according to their respective molecular weight M_w and intrinsic viscosity values, as well as the intermediate character of the two active but narrow-spectrum activity molecules (PnPS18C and PnPS12F).

Supplementary Figure S6. Classification of the tested polysaccharides according to their molecular weight (M_w) and intrinsic viscosity [η] as indicated in Table 1.

In figure S4 reference is made to several active polysaccharides. In reality these are 4 polysaccharides, the caption must be corrected.

We agree with the reviewer, and we corrected the caption of Figure S4 accordingly.

Some typos are scattered throughout the text or figures.

We carefully went through the text and Figures to correct the remaining typos pointed by the reviewer.

Reviewer #2 (Remarks for the Author):

This study offers biophysical and elektrokinetic methods to identify new surface-active polysaccharides for antibiofilm activity. The work is significant for antibiofilm activity.

The antibiofilm activity is poorly studied and it is not clear if the authors studied a biofilm or planktonic bacteria from the method they used, see also comments below. Therefore it is suggested to add a better antibiofilm assay comparable as in the paper of Valle et al. 2006. Additional antibiofilm methods are necessary to proof the antibiofilm activity of the isolated polysaccharides.

The reviewer has a good point, and we are now providing results obtained with the requested additional biofilm method, which shows that the most active broad-spectrum polysaccharide Vi strongly inhibits *E. coli* and *S. aureus* biofilm formation in continuous flow biofilm microfermentors (see new Figure 3, pasted below). This inhibition is not detected with treatment by the inactive polysaccharide PnPS8. Moreover, use of continuous flow biofilm microfermentors confirm the narrow-spectrum activity of PnPS18C polysaccharide (active on *S. aureus* but inactive on *E. coli*) and of PnPS12F (inactive on *S. aureus* but active on *E. coli*), as shown in the new Figure 3).

These results are therefore in full agreement with the conclusions reported in our original manuscript. They are now presented in the revised manuscript (new Figure 3) and specified in the revised manuscript as follows.

Page 7 Line 175-178: Result section

“Using static microtiter plate biofilm assay followed by crystal violet staining, we showed that, at equivalent concentration (100µg/mL), Vi, MenA and MenC polysaccharides were as active as G2cps in inhibiting *E. coli* biofilm formation but were less active on *S. aureus* biofilms (Figure 2A, B).”

And

Page 7 Line 183-189: Result section

“We confirmed these results using a dynamic assay using continuous flow biofilm microfermentors and showed that Vi also strongly inhibited *E. coli* and *S. aureus* biofilm formation, while this inhibition was not observed with treatment by the inactive polysaccharide PnPS8 (Figure 3). This analysis also confirmed the narrow-spectrum activity of PnPS18C polysaccharide (active on *S. aureus* but inactive on *E. coli*) and of PnPS12F (inactive on *S. aureus* but active on *E. coli*) (Figure 3).”

New Figure 3. Biofilm inhibitory effect of polysaccharides in continuous flow biofilm microfermentors.

The question is how do you know you have a biofilm and not just a planktonic culture? Normally when you have a biofilm you first need to adhere bacteria to a surface from growth media or buffer, wash, add growth media and let the biofilm grow for the appropriate time. You need to add the quantification assay.

The reviewer's description of a biofilm assay could indeed correspond to a dynamic biofilm assay under flow conditions (such an assay *is now present in the revised manuscript, see points above and further response below*).

It does not, however, correctly described the crystal violet-based biofilm assay used in the initially submitted version. In this valid and widely used assay, after a low inoculation level (0,02 OD_{600nm}) both planktonic and biofilm cells indeed coexist until wash steps eliminate planktonic bacteria, and crystal violet staining allows the quantification of the biofilm biomass only, which is attached to the well surfaces. This time-tested assay also enables quantitative analyses that are systematically presented in the study (e.g., Figure 1, 2, S2, S4, S6)

Nevertheless, we now also provide biofilm analysis in dynamic biofilm assay using continuous flow biofilm microfermentors to further address the reviewer's concern. In this assay, flow allows the biofilm to develop while planktonic bacteria are continuously eliminated (see the new Figure 3 pasted in our response to previous referee's comment). Although this system does not allow to test as many polysaccharides than the microtiter plate assay, both static and dynamic biofilm assays are in full agreement regarding non-activity, broad activity or selected narrow-spectrum activity of the tested polysaccharides, which supports and strengthens the conclusions reported in our original manuscript.

An improved complete description of these two biofilm assays adopted in the context of our study is now introduced in the Material and methods section as follows:

Page 16-17 Line 461-493: **Method section**

“Biofilm formation and inhibition assays

Static tests: microtiter plate crystal violet biofilm assay

Overnight cultures were adjusted to an OD_{600nm} of 0.02 before inoculating 50 µl into 96-well polyvinyl chloride (PVC) plates (Falcon; Becton Dickinson Labware, Oxnard, CA) and added at a 1:1 ratio to 50 µl of filter-sterilized purified polysaccharide at the desired concentration: i) 50 µg/ml for standardized antibiofilm activity experiments; ii) 3.125-100 µg/ml to test the level of activity of the macromolecules. Biofilms were left to grow for 16 h at 37°C. The supernatant was then removed from each well, washed three times with water and biofilm was stained with 1% crystal violet solution for 15 min. After removal of the crystal violet solution, biofilms were washed three times with water. For quantification of biofilm formation, dried stained biofilms were resuspended in 30% acetic acid and absorbance was measured at 585 nm using an infinite Tecan infinite M200 PRO plate reader. Biofilm formation with different strains is represented in values normalized to average biofilm formation with the control strains.

Dynamic tests: continuous flow biofilm microfermentor assay

Sterile continuous-flow biofilm microfermentors containing a removable glass spatula were used as described in (Ghigo 2001) (see also <https://research.pasteur.fr/en/tool/biofilm-microfermenters/>). Medium flow was adjusted to 60 ml/h with internal bubbling agitation with filter-sterilized compressed air. This flow rate minimizes planktonic growth over biofilm development by constant dilution of non-biofilm bacteria. Inoculation was performed by dipping the glass spatula for 5 min in OD₆₀₀=1 M63B1 0.4% glucose minimal medium (M63B1glu) for *E. coli* strain and in tryptic soy broth 0,25% glucose (TSBglu) for *S. aureus* strain, supplemented or not with 100 µg/ml purified polysaccharide Vi, PnPS8, PnPS18C, PnPS12F. After bacteria were allowed to attach to the spatulas, they were then reintroduced into the microfermentor filled with the appropriate media supplemented also with the same final concentration of tested polysaccharide. After 24 h of continuous culture, pictures of the glass spatula were taken, and biofilm biomass estimated by determining the OD_{600nm} of the biofilm resuspended from the internal microfermentor glass slide.”

Methods, in which medium where *S. aureus* and *S. epidermidis* cultured, which cultures get 0.25 and 0.50% glucose? Can you give a bit more details?

As indicated in the Material and Methods section, all Gram + bacteria, including *S. aureus* and *S. epidermidis*, were grown in TSB supplemented with 0.25% glucose. *Enterobacter cloacae* was cultivated in TSB supplemented with 0.5 % glucose. However, it was indeed not mentioned that this bacterium is a Gram-negative bacterium, and the corresponding section was indeed misleading.

We have now corrected and specified explicitly all adopted cell growth conditions in the revised manuscript. The added corrections read as follows:

Page 17 Line 448-453: **Method section**

“Gram- bacteria were grown in 0.4% glucose-M63B1 minimal medium (M63B1) or in lysogeny broth (LB) medium at 37°C, with appropriate antibiotics when required. Gram+ strains were cultured in tryptic soy broth (TSB) supplemented with 0.25% or 0.5% glucose (*E. cloacae*). Potential biocidal effect of antibiofilm polysaccharides was evaluated from growth curve determination in the presence of 100 µg/ml of purified polysaccharide.”

Specify cold centrifugation.

This is now specified.

Page 18 Line 513: **Method section**

“After cold (4°C) centrifugation at 7500 rpm for 10 minutes, supernatants were filtered through a 0.22µm filter.”

Check if all abbreviations are introduced.

We carefully checked for all abbreviations, and we have now removed all typos.

The assay used for non-biocidal activity is lacking in the Methods, it is partly described in the caption, but growth media for example is lacking. Also add the agitation speed.

We agree, and we have now added the requested experimental details on non-biocidal activity determination as well as on agitation speed. The corrected text reads as follows:

Page 17 Line 494-499: **Method section**

Test of non-biocidal activity

Growth curves of *S. aureus* and *E. coli* exposed to 100 µg/ml were measured in the presence of each indicated tested macromolecule (Figure S3). The bacterial strains were inoculated in microtiter plates at OD_{600nm} of 0.05 and let to grow in TECAN plate reader (Infinite 200 Pro) with a 2 mm orbital shaking amplitude for 24 h at 37°C. Bacterial growth was determined by the TECAN plate reader.”

Can you use this on all surfaces? So far you tested it only on PVC?

As the objective of our study was to identify intrinsic physical-chemical properties of polysaccharides that determine their anti-biofilm activity, we chose to focus our experimental work on a broad range of polysaccharides rather than on different cell adhesion surfaces. However, in addition to the PVC surface used in the microtiter plate biofilm assay presented in our original manuscript, we now also provide

biofilm formation results obtained on a glass surface, which corresponds to the surface of the spatula inserted in the biofilm microfermentor (see new Fig 3). These latter results on glass confirmed the broad-spectrum activity of Vi and the narrow-spectrum activity of PnPS18C/12F, which fully supports the results obtained with standard microtiter biofilm *in vitro* assays.

In addition, we also presented data showing that the coating of hydrophilic glass and hydrophobic polyester plastics by the tested polysaccharides did not significantly modify their wettability (see Figure S7).

Reviewer #3 (Remarks for the Author):

This manuscript is well written and organized. The authors have studied the biophysical properties of non-biocidal antibiofilm polysaccharides and the correlation between the properties and the antibiofilm activity in order to examine new approaches to engineering and screening of antibiofilm polysaccharides. The methods are generally appropriate to fulfill the overall aim of the study, but some conclusions drawn in the result section should be carefully revised to be more specific and take into consideration the study limitations and the sample size. Specific comments follow. The authors make a systematic contribution to the research literature in this area of investigation. Overall, this is a good-quality manuscript with interesting new highlights on the biophysical properties shared by polysaccharides with antibiofilm activity.

Thank you

ABSTRACT:

Page 2, Line 11-13

“we screened a collection of 32 purified capsular polysaccharides and identified seven new compounds with non-biocidal activity against biofilms formed by *Escherichia coli* and/or *Staphylococcus aureus*.”

In the section page 4 line 25 and page 6 line 31, The compounds are described as previously known. Therefore, the statement “seven new compounds” as an “novel” should be revised or clarified to reflect that the antibiofilm activity is the new finding. Please, revise this statement throughout the paper.

The reviewer has a good point, and we modified the manuscript as follows.

Page 2 Line 50: *Abstract*

“and identified seven compounds with non-biocidal activity against biofilms.”

Page 4 Line 127: *Introduction*

“Among those, we identified nine new non-biocidal polysaccharides inhibiting biofilm formation by prototypical nosocomial pathogens.”

Page 6 Line 164 *Result section title*

“Screening a collection of bacterial capsular polysaccharides reveals compounds with non-biocidal antibiofilm activity”

See also Page 13 Line 351 *Discussion*

« In this study, we identified nine non-biocidal antibiofilm macromolecules among a collection of 31 different high molecular-weight bacterial capsular polysaccharides of known structure and composition.”

“the polysaccharide mobility”. Mobility of all polysaccharides was assessed

We indeed did not test them all and we agree that this should be more clearly indicated.

Accordingly, we replaced the following original sentence in the abstract section Line 51-53 by:

“We analyzed the polysaccharide mobility under applied electric field conditions and showed that active and inactive polysaccharide polymers display distinct electrokinetic properties and that all active macromolecules shared high intrinsic viscosity features.”

By

Page 2 Line 51-55: *Abstract*

“We measured and theoretically interpreted the electrophoretic mobility under applied electric field conditions of a subset of 21 active and inactive capsular polysaccharides and showed that active and inactive polysaccharide polymers display distinct electrokinetic properties and that all active macromolecules shared high intrinsic viscosity features.”

Page 2, Line 17-18

“Based on these characteristics, we identified two additional antibiofilm capsular polysaccharides with high density of electrostatic charges and their permeability to fluid flow.”

distinct electrokinetic properties

Thank you: We clarified the sentence as follows

We replaced the following initial sentence in the *Abstract*

“Based on these characteristics, we identified two additional antibiofilm capsular polysaccharides with high density of electrostatic charges and their permeability to fluid flow.”

By

Page 2 Line 55-59: *Abstract*

“Despite the lack of specific molecular motif associated with antibiofilm properties, the search for capsular polysaccharides with high density of electrostatic charges and permeability to fluid flow enabled us to identify two additional molecules with broad-spectrum antibiofilm activity.”

We also changed “The characterization of a specific electrokinetic signature...”

for :

Page 2 Line 61: *Abstract*

“The characterization of a distinct electrokinetic signature”

Page 2, Line 21-23

“This characterization of a specific electrokinetic signature for polysaccharides displaying antibiofilm activity opens new perspectives to identify or engineer non-biocidal surface-active macromolecules to control biofilm formation in medical and industrial settings.”

We simplified this sentence as follows

Page 2 Line 6-64: *Abstract*

“The characterization of a distinct electrokinetic signature associated with polysaccharides displaying antibiofilm activity opens new perspectives to identify or engineer non-biocidal surface-active macromolecules to control biofilm formation in medical and industrial settings”

INTRODUCTION

Page 3, Line 4

“negatively impact human activities”. What is meant by activities? Biofilm developing on medical or industrial surfaces affect human health.

We consider that limiting the negative impact of biofilm formation to health would be too restrictive. Biofilm formation in industrial settings indeed clogs pipes, promotes corrosion or reduces flow. This (non-exhaustive) list of adverse biofilm-related impacts are not related to health, but they are still of major concern. That is why we motivate our use of the broader terminology ‘impact on human activities’.

Page 3, Line 12-14

“These approaches are limited by the rapid masking of the coated surfaces by bacterial and organic debris, and are associated with worrisome selection of resistance upon repeated contact with treated surfaces”. This sentence needs to be reworded. It is not clear if it is bacterial debris or actual bacteria and organic debris, the resistance is not specified as antibiotic resistance, and word masking doesn’t seem to be used appropriately. Also, what does “upon repeated contact refers to?

We replaced the following initial sentence

“These approaches are limited by the rapid masking of the coated surfaces by bacterial and organic debris, and are associated with worrisome selection of resistance upon repeated contact with treated surfaces (Danese 2002).”

clarified this sentence as follows

Page 3 Line 78-82: *Introduction*

“These biocidal approaches are limited by the rapid accumulation of dead bacteria and organic debris, which reduces the activity of the coated surfaces towards new incoming cells. Moreover, the use of surfaces releasing biocides such as antibiotics is associated with worrisome selection of antibiotic resistance (Danese 2002).”

Page 3, Line 22:

“to protect patient care equipment” is unclear as to what type of protection. Consider removing or clarifying. Suggestion: “to protect patient care equipment from pathogen colonization and...”

We agree with the reviewer’s suggestion and corrected the text accordingly.

Page 3 Line 90: *Introduction*

“...has been proposed to constitute an effective solution to protect patient care equipment from pathogen colonization.”

Page 3, Line 25

“bio-sourced strategies”. Do you mean biobased strategies?

Page 3, Line 25-27

“preventing and/or disrupting biofilms based on the use of quorum sensing inhibitors”. Suggested rewording the sentence as “preventing and/or disrupting biofilm formation using quorum sensing inhibitors”

Yes, thank you. We corrected accordingly.

Page 3 Line 93-95: **Introduction**

“Non-biocidal and **bio-based** strategies are also actively explored, including approaches preventing and/or disrupting biofilms **using** quorum sensing inhibitors that interfere with bacterial communications (Rendueles and Ghigo 2012)”

RESULTS

Page 7, Line 17

statement title “Polysaccharide conformation and high intrinsic viscosity are predictive indicators of antibiofilm activity” is an overstatement in the result section and should be revised as it is “a potentially predictive indicator” at this point.

We disagree on this point as this section precisely describes how “high intrinsic viscosity *could be indicative* of a potential antibiofilm activity” (Line 228) and we then proceeded with the screening of molecules displaying the characteristics highlighted by Figure 4, and these molecules turned out to display the expected antibiofilm activity (MenY and MenW135).

Page 7, Line 22, Page 10, Line 2, and other result sections:

“and inactive (PnPS1, PnPS8, PnPS19A, PnPS9V, PnPS7F, PnPS22F and PnPS14) polysaccharides by HPSEC”. “compared to those of seven inactive polysaccharides (PnPS1, PnPS9V, PnPS8, PnPS19A, PnPS7F, PnPS22F and PnPS14)”

Why and how these specific inactive polysaccharides were selected to compare their properties to active compounds since 32 Polysaccharides were screened and at least 23 were inactive.

The inactive polysaccharides were chosen so as to form a subset of inactive capsules displaying different structures and compositions, as detailed in Table S1.

Have all inactive compounds been tested for their intrinsic viscosity and for their electrokinetic behavior?

Although we performed additional electrokinetic analysis to examine all 10 active polysaccharides, and although we added inactive polysaccharides (11 in total now) to our list of macromolecules subjected to electrokinetic analysis, we could not examine all inactive capsules due to logistical reasons. However, the reviewer makes a good point and we have therefore systematically determined the molecular weight and intrinsic viscosity of all 31 tested polysaccharides which are now presented in the revised version of Table 1.

These data are now plotted in a more complete version of revised Figure S5 pasted below, confirming the clear separation between active and inactive molecules according to molecular weight Mw and intrinsic viscosity, as well as the intermediate character of the two active but narrow-spectrum activity molecules (18C and 12F), therefore strengthening the conclusions drawn in the originally submitted manuscript.

Supplementary Figure S5. Classification of the tested polysaccharides according to their molecular weight (M_w) and intrinsic viscosity $[\eta]$ as indicated in Table 1.

Page 8, line14

“These results indicated that specific polysaccharide conformation, reflected by a high intrinsic viscosity (Yang 15 1961), is a determinant of antibiofilm activity.” This study showed that high intrinsic viscosity can be a “predictive indicator of antibiofilm activity”. However, because of the sample size and other properties evaluated, the concluding sentence(is a determinant of antibiofilm activity) of this paragraph appears to be an overstatement.

We agree and toned down this sentence as follows

Page 8 Line 226-228: *Introduction*

“These results indicated that specific polysaccharide conformation, reflected by a high intrinsic viscosity (Yang 1961), could be a determinant of antibiofilm activity.”

REVIEWER COMMENTS

Reviewer #2 (Remarks to the Author):

The authors have improved the manuscript by including many of the reviewers comments and if they disagreed they gave a clear response to the issue from the reviewer.

Reviewer #3 (Remarks to the Author):

Reviewer#1 initial comments and new comments:

Initial reviewer's comments: The authors present a manuscript in which they highlight that bacterial capsular polysaccharides owe their antibiofilm potential to structural requirements as high intrinsic viscosity and specific electrokinetic signature. The results presented are very interesting and could reasonably be useful for the research and production of new antibiofilm agents. Also, the methodologies used seem appropriate as well as the observations that the authors have obtained and the way in which they have reworked them in the conclusions. The methods are well described, and the equalizations used are also, well-motivated. On the basis of this premise, I believe that the work, however, needs an adequate review in some points that are ambiguous or even incorrect.

1. In detail, I find the description of the number of capsules used in certain experiments very confusing. On some points as several times, I have not found a match between the number indicated in the text and the one actually indicated in the figures.

Initial reviewer's comments: For example, on page 4 line 25, the authors plan to use 32 capsules purified by both Gram+ and Gram-bacterial strains. This number is never respected in any figure or table in the manuscript.

 This concern was addressed in the rebuttal document and the manuscript.

Initial reviewer's comments: On page 6 line 29, the authors declare that they screened 30 different high molecular-weight bacterial polysaccharides to detect properties similar to that G2cps, the latter used as a reference. Thus, the capsules should be 31 and this number is correctly reported in table S1. However, when the reader analyzes Figure 2, the capsules indicated in the graphs are 29 including G2cps. Authors should better specify or correct.

 This concern was addressed in the rebuttal document and the manuscript.

Initial reviewer's comments: Further clarifications are needed regarding the motivation for choosing 4 capsules and not 5 in the experiment described on page 7, line 10, and shown in figure S4.

 This concern was not addressed specifically in the rebuttal document.

 The reviewer was asking why MenC was not tested for antibiofilm spectrum of activity along with Vi, MenA, PRP, and PnPS3 on the same panel of biofilm-forming Gram+ or Gram-bacteria (activity of the 4 capsules is shown in figure S4). Figure S4 in the revised version still only has 4 graphs for Vi, MenA, PRP, and PnPS3 activity and this is the experiment described on page 7, line 10 of the previous version, and is still the same in the new version line 191-195.

 The author provided a rationale and stated "we first did not include MenC because MenA and MenC displayed quite similar primary structure, molecular weight Mw, and intrinsic viscosity. However, the reviewer has a good point and we now have addressed this issue..."

But instead, in this section of the rebuttal, the author provided a clear rationale for not testing all 5 active capsules for their electrokinetic properties (electrophoretic mobility) and performed additional experiments to include MenC in the electrokinetic analysis and the updated data was provided in form

of new graphs replacing the previous ones (figure 5 and 6, previously 4 and 5). However, this was not the reviewer's concern in this section.

Initial reviewer's comments: Another apparent contradiction concerns the different number of capsules reported in the tables. Table 1 shows 17 capsules, table S2 shows 15. In the text, this difference should be better justified if supported by real reasons.

 This concern was addressed in the rebuttal document and the manuscript. The author provided a clear rationale for the different numbers of capsules reported in the tables. The author provided additional data for both table 1 and table S2 with a clear rationale as well as an updated version of Figure S6 including all tested polysaccharides (formerly S5).

Initial reviewer's comments: In figure S4 reference is made to several active polysaccharides. In reality, these are 4 polysaccharides, the caption must be corrected.

 This concern was addressed in the rebuttal document and the supplementary materials document. The caption for figure S4 in the revised version is now "Spectrum of activity of 4 active polysaccharides"

Initial reviewer's comments: Some typos are scattered throughout the text or figures.

 This concern was addressed in the rebuttal document and many typos were corrected in the manuscript. However, I see the following in line 321 page 11: the word tested is repeated twice and one of them should be removed.

The editorial proofreading can address any remaining typos if any.

 A few minor additional points:

 Additional concern: Figure 4 (previously figure 3). The title states "MenA and MenY antibiofilm activity" but this figure is referring to "Antibiofilm activity of MenY and MenW135 in comparison with previously identified active macromolecules (Vi and MenA)". My understanding is that the caption should be "MenY and MenW135 antibiofilm activity"

 Additional concern: in line 276 it states: "Solution (1 mM to 100 mM range) for all 10 broad spectrum (Vi, MenA, MenC, MenY, MenW135, G2cps, PRP, PnPS3). I believe it should state "all 8 broad spectrum". 10 were active in this study but only 8 were BROAD spectrum. Suggested rewording: "for all, 8 broad spectrum (Vi, MenA, MenC, MenY, MenW135, G2cps, PRP, PnPS3) and 2 narrow spectrum (PnPS18C and PnPS12F) antibiofilm polysaccharides."

 Additional concern: line 320; the use of the word "excellent" is subjective and it is suggested to be replaced or removed.

 Additional concern: line 223; it is stated "MenW135, presenting high intrinsic viscosity (Table 1 and Supplementary Figures S3 and S5)". Figure S3 should be removed from the parenthesis because it is about "The identified antibiofilm polysaccharides are non-biocidal" and not the "high intrinsic viscosity"

• **Reviewer #2** (Remarks to the Author):

The authors have improved the manuscript by including many of the reviewers comments and if they disagreed they gave a clear response to the issue from the reviewer.

We thank reviewer #2 for these comments.

• **Reviewer #1** initial comments reviewed by Reviewer 3 based on our responses after 1st round of review (see previous rebuttal):

Initial reviewer's 1 comments: *The authors present a manuscript in which they highlight that bacterial capsular polysaccharides owe their antibiofilm potential to structural requirements as high intrinsic viscosity and specific electrokinetic signature. The results presented are very interesting and could reasonably be useful for the research and production of new antibiofilm agents. Also, the methodologies used seem appropriate as well as the observations that the authors have obtained and the way in which they have reworked them in the conclusions. The methods are well described and the equalizations used are also well motivated. On the basis of this premise, I believe that the work, however, needs an adequate review in some points that are ambiguous or even incorrect.*

In detail, I find the description of the number of capsules used in certain experiments very confusing in some points as several times I have not found a match between the number indicated in the text and the one actually indicated in the figures.

For example, on page 4 line 25, the authors plan to use 32 capsules purified by both Gram+ and Gram-bacterial strains. This number is never respected in any figure or table in the manuscript.

=> **Reviewer 3 assessment of our response:**

This concern was addressed in the rebuttal document and the manuscript.

Initial reviewer's 1 comments: *On page 6 line 29, the authors declare that they screened 30 different high molecular-weight bacterial polysaccharides to detect properties similar to that G2cps, the latter used as a reference. Thus, the capsules should be 31 and this number is correctly reported in table S1. However, when the reader analyzes Figure 2, the capsules indicated in the graphs are 29 including G2cps. Authors should better specify or correct.*

=> **Reviewer 3 assessment of our response:**

This concern was addressed in the rebuttal document and the manuscript.

Initial reviewer's 1 comments: *Further clarifications are needed regarding the motivation for choosing 4 capsules and not 5 in the experiment described on page 7, line 10 and shown in figure S4.*

=> **Reviewer 3 assessment of our response:**

This concern was not addressed specifically in the rebuttal document.

As indicated in our previous response, and as further correctly pointed by reviewer 3, we did provide a rationale for not including MenC in the analysis

“we did not include MenC because MenA and MenC displayed quite similar primary structure, molecular weight Mw, and intrinsic viscosity.” However, we now realize that this justification may not be sufficient as it does not directly address the spectrum of menC antibiofilm activity. Accordingly, we chose to

perform additional experiments so as to test the spectrum of activity of MenC: these new data are now reported in the revised version of the manuscript as a new supplementary Figure S4, where the antibiofilm activity of all 5 active capsules is provided, including the one of MenC capsule that was missing in our previous version of the manuscript (see New Fig S4 pasted below).

Figure S4. Spectrum of activity of 5 active antibiofilm polysaccharides.

Initial reviewer's 1 comments: Another apparent contradiction concerns the different number of capsules reported in the tables. Table 1 shows 17 capsules, table S2 shows 15. In the text, this difference should be better justified if supported by real reasons.

=> **Reviewer 3 assessment of our response:**

This concern was addressed in the rebuttal document and the manuscript. The author provided a clear rationale for the different numbers of capsules reported in the tables. The author provided additional data for both table 1 and table S2 with a clear rationale as well as an updated version of Figure S6 including all tested polysaccharides (formerly S5).

Thank you.

Initial reviewer's 1 comments: In figure S4 reference is made to several active polysaccharides. In reality these are 4 polysaccharides, the caption must be corrected.

=> **Reviewer 3 assessment of our response**

This concern was addressed in the rebuttal document and the supplementary materials document. The caption for figure S4 in the revised version is now "Spectrum of activity of 4 active polysaccharides"

Given that we have now included MenC in the analysis (cf. revised Figure S4, pasted above), the caption of Figure S4 now reads as "Spectrum of activity of 5 active antibiofilm polysaccharides."

Some typos are scattered throughout the text or figures.

=> **Reviewer 3 assessment of our response**

This concern was addressed in the rebuttal document and many typos were corrected in the manuscript.

• **Reviewer #3** (Remarks to the Author):

I see the following in line 321 page 11: the word tested is repeated twice and one of them should be removed.

We thank reviewer 3 for spotting this. The extra "tested" has been removed.

Additional concern: Figure 4 (previously figure 3). The title states "MenA and MenY antibiofilm activity" but this figure is referring to "Antibiofilm activity of MenY and MenW135 in comparison with previously identified active macromolecules (Vi and MenA)". My understanding is that the caption should be "MenY and MenW135 antibiofilm activity"

The reviewer is right and we have changed the caption of Figure 4 accordingly:

"Figure 4. *MenY and MenW135 antibiofilm activity*"

Additional concern: in line 276 it states: "Solution (1 mM to 100 mM range) for all 10 broad spectrum (Vi, MenA, MenC, MenY, MenW135, G2cps, PRP, PnPS3). I believe it should state "all 8 broad spectrum". 10 were active in this study but only 8 were BROAD spectrum. Suggested rewording: "for all, 8 broad spectrum (Vi, MenA, MenC, MenY, MenW135, G2cps, PRP, PnPS3) and 2 narrow spectrum (PnPS18C and PnPS12F) antibiofilm polysaccharides."

We agree and changed the text as follows:

"...we performed blind measurements of the electrophoretic mobility (μ) as a function of NaNO₃ electrolyte concentration in solution (1 mM to 100 mM range) for all 8 broad spectrum (Vi, MenA, MenC, MenY, MenW135, G2cps, PRP, PnPS3) and 2 narrow spectrum (PnPS18C and PnPS12F) polysaccharides."

Additional concern: line 320; the use of the word “excellent” is subjective and it is suggested to be replaced or removed.

We agree and changed the text from:

“This quantitative analysis highlighted an excellent reconstruction of the electrophoresis data measured for all tested macromolecules (Figure 5)”

To:

“This quantitative analysis highlighted a proper theoretical reconstruction of the electrophoresis data measured for all tested macromolecules (Figure 5)”

Additional concern: line 223; it is stated “MenW135, presenting high intrinsic viscosity (Table 1 and Supplementary Figures S3 and S5”. Figure S3 should be removed from the parenthesis because it is about “The identified antibiofilm polysaccharides are non-biocidal” and not the “high intrinsic viscosity”

We agree and moved up the reference to Figure S3 (growth curve) right after the antibiofilm mention of non-biocidal activity as follows

“...and we identified two such non-biocidal polysaccharides, MenY and MenW135 (Supplementary Figures S3), presenting high intrinsic viscosity (Table 1 and supplementary Figure S5).”

REVIEWERS' COMMENTS

Reviewer #3 (Remarks to the Author):

The researchers systematically addressed all the concerns of the reviewer by performing additional experiments, making the suggested changes, and including the reviewer's comments.

The reviewer has no more comments.

Thank you!